# A human antibody against pathologic IAPP aggregates protects beta cells in type 2 diabetes models

Fabian Wirth [1,8], Fabrice D. Heitz[1,8], Christine Seeger[1], Ioana Combaluzier[1], Karin Breu[1], Heather C. Denroche [2], Julien Thevenet [3], Melania Osto[4], Paolo Arosio [5], Julie Kerr-Conte[3], C. Bruce Verchere[2], François Pattou [3], Thomas A. Lutz [4], Marc Y. Donath [6], Christoph Hock[1,7], Roger M. Nitsch[1,7] & Jan Grimm [1] ✉

In patients with type 2 diabetes, pancreatic beta cells progressively degenerate and gradually lose their ability to produce insulin and regulate blood glucose. Beta cell dysfunction and loss is associated with an accumulation of aggregated forms of islet amyloid polypeptide (IAPP) consisting of soluble pre-fibrillar IAPP oligomers as well as insoluble IAPP fibrils in pancreatic islets. Here, we describe a human monoclonal antibody selectively targeting IAPP oligomers and neutralizing IAPP aggregate toxicity by preventing membrane disruption and apoptosis in vitro. Antibody treatment in male rats and mice transgenic for human IAPP, and human islet-engrafted mouse models of type 2 diabetes triggers clearance of IAPP oligomers resulting in beta cell protection and improved glucose control. These results provide new evidence for the pathological role of IAPP oligomers and suggest that antibody-mediated removal of IAPP oligomers could be a pharmaceutical strategy to support beta cell function in type 2 diabetes.

Type 2 diabetes (T2D) is a chronic metabolic disorder characterized by insulin resistance and progressive dysfunction and loss of insulin-producing pancreatic beta cells, resulting in insulin deficiency and elevated blood glucose. Beta cell death is accompanied by the accumulation and aggregation of the 37-residue peptide hormone islet amyloid polypeptide (IAPP or amylin) that is co-secreted with insulin and forms islet amyloid found in the majority of people with T2D patients[1]. In its physiological monomeric conformation, IAPP acts as a regulator of glucose homeostasis through satiety control and inhibition of gastric emptying[2,3]. As IAPP is highly amyloidogenic, under

conditions of increased secretory demand it can readily misfold and aggregate into soluble oligomers and insoluble amyloid fibrils that are thought to contribute to beta cell dysfunction and death in T2D[4–6]. The involvement of IAPP aggregates in beta cell decline and T2D progression is supported by an increasing body of evidence. First, amyloid severity inversely correlates with beta cell area in pancreatic islets of T2D patients[4]. Second, a sporadic mutation in the human *IAPP* coding sequence leading to a S20G amino acid substitution is associated with a higher propensity for aggregation, an increased risk for developing T2D and a more severe form of the disease[7–9]. Third, cats and primates

[1]Neurimmune AG, Wagistrasse 18, 8952 Schlieren, Switzerland. [2]BC Children's Hospital Research Institute and Centre for Molecular Medicine and Therapeutics, Departments of Surgery and Pathology & Laboratory Medicine, University of British Columbia, A4-151 950 W 28 Ave, Vancouver, BC, Canada. [3]Univ-Lille, Inserm, CHU Lille, U1190 - EGID, F-59000 Lille, France. [4]Institute of Veterinary Physiology, Vetsuisse Faculty of the University of Zürich, Winterthurerstrasse 260, 8057 Zürich, Switzerland. [5]Institute for Chemical and Bioengineering, ETH Zürich, Vladimir-Prelog-Weg 1-5/10, 8093 Zürich, Switzerland. [6]Clinic for Endocrinology, Diabetes & Metabolism, and Department of Biomedicine, University Hospital Basel, Hebelstrasse 20, 4031 Basel, Switzerland. [7]Institute for Regenerative Medicine-IREM, University of Zürich, Wagistrasse 12, 8952 Schlieren, Switzerland. [8]These authors contributed equally: Fabian Wirth, Fabrice D. Heitz. ✉e-mail: jan.grimm@neurimmune.com

producing amyloidogenic variants of IAPP and forming pancreatic islet amyloid deposits naturally develop signs of T2D[10]. In non-human primates, islet amyloid severity was also shown to correlate with beta cell loss and T2D progression[11,12]. Fourth, transgenic mice and rats expressing the amyloidogenic human IAPP (hIAPP) spontaneously develop a T2D phenotype characterized by islet amyloidosis and decreased beta cell mass[13–16]. Development of T2D in wildtype mice harboring the non-amyloidogenic rodent IAPP (rIAPP) is absent, but can be triggered by high-fat diet paradigms, β-cell selective toxins, or by introducing spontaneous variants in genes other than IAPP such as leptin or insulin[17]. Furthermore, humanization of the non-amyloidogenic porcine IAPP using CRISPR/Cas9 gene editing leads to T2D in miniature pigs[18]. Of note, IAPP aggregation is also linked to beta cell deterioration in human islets cultured in high glucose medium or transplanted into mice or humans with type 1 diabetes[19–21]. Synthetic IAPP aggregates, primarily oligomers produced at an early stage of amyloid fibril formation, induce beta cell dysfunction and apoptosis in vitro[22]. Their cytotoxicity presumably results from membrane permeabilization[23,24], induction of oxidative and ER stress[25,26], and pro-inflammatory cytokine release[27,28]. However, the contribution of IAPP oligomers to beta cell loss and T2D progression remains controversial and has not been yet clearly established in vivo. Here, we describe a human monoclonal antibody α-IAPP-O selectively targeting IAPP oligomers in T2D. α-IAPP-O neutralizes IAPP aggregate toxicity by preventing membrane disruption and apoptosis in vitro. In male rats and mice transgenic for human IAPP, and human islet-engrafted mouse models of T2D, α-IAPP-O triggers clearance of IAPP oligomers resulting in beta cell protection and improved glucose control. These data support the toxic role of IAPP oligomers in T2D and provide a new avenue for beta cell protective therapies using α-IAPP-O.

## Results

### A human monoclonal antibody with high affinity and selectively for aggregated hIAPP

We identified, cloned and recombinantly expressed a monoclonal antibody of human IgG1 subclass (termed α-IAPP-O) selectively targeting pathologic hIAPP aggregates by analyses of complements of human memory B cells derived from a clinically selected human population composed of healthy elderly donors. α-IAPP-O selectively immunoreacted at a low nanomolar concentration with extracellular hIAPP aggregates present on amyloid-positive pancreatic islets from type 2 diabetic subjects, with absence of binding to native hIAPP within insulin-producing pancreatic beta cells (Fig. 1a, b and Supplementary Fig. 1a). α-IAPP-O's high target specificity was further confirmed by absent off-target binding towards more than 6000 native human secreted and membrane proteins, including physiological hIAPP monomer (Supplementary Fig. 1b–e), as well as lack of binding towards unrelated disease-associated amyloidogenic proteins and β-amyloid plaques in the brain of Alzheimer's disease patients (Supplementary Fig. 2). α-IAPP-O binding kinetics revealed strong binding of hIAPP aggregates ($K_D = 1.54$ nM) likely consisting of a heterogenous mixture of aggregated species, as compared to monomeric hIAPP (biotin-hIAPP, $K_D = 2.84\,\mu$M) using biolayer interferometry (Fig. 1c, d and Supplementary Table 1). Binding affinity of the monovalent α-IAPP-O Fab fragment to hIAPP aggregates ($K_D = 6.47\,\mu$M) was 4000-fold lower compared to the bivalent IgG1 format (Fig. 1e and Supplementary Table 1), suggesting a strong avidity component in α-IAPP-O binding to aggregated forms of hIAPP. Similar findings have been described for therapeutic antibody candidates generated from human B-cell repertoires against other disease-causing protein aggregates such as β-amyloid and α-synuclein[29,30]. The α-IAPP-O epitope was mapped to an N-terminal sequence conserved among hIAPP and less amyloidogenic IAPP orthologues and binding was impaired by proline substitution in the amyloidogenic region responsible for amyloid fibril formation,

C-terminal truncation, and SDS-induced denaturation (Supplementary Fig. 3).

### α-IAPP-O preferentially binds to hIAPP prefibrillar oligomers

Human IAPP rapidly aggregated into amyloid fibrils visualized by transmission electron microscopy, and amyloid growth monitored by thioflavin-T (ThioT) fluorescence followed a sigmoidal kinetics characterized by a lag phase, an exponential growth phase and an equilibrium phase (Fig. 2a), as described earlier by different techniques[31–33]. To elucidate the nature of aggregated hIAPP species recognized by α-IAPP-O, we performed time-resolved immunoblot analysis of fractions collected over the course of amyloid fibril formation. α-IAPP-O was shown to preferentially immunoreact with transient prefibrillar oligomers that are produced during the lag and the growth phases of amyloid formation (Fig. 2b). In contrast to a non-selective IAPP antibody (α-IAPP), α-IAPP-O bound neither to monomeric hIAPP nor to non-amyloidogenic rodent IAPP (rIAPP) and detergent-denatured hIAPP aggregates. α-IAPP-O recognized high molecular weight hIAPP species (>200 kDa) but not monomers or chemically cross-linked aggregates generated in vitro and resolved by non-reducing SDS-PAGE and Western blotting (Fig. 2c). We next studied the effect of α-IAPP-O on hIAPP aggregation using kinetic modeling of molecular events underlying amyloid fibril formation[34–38]. Using ThioT fluorescence measurement, quantitative kinetic analysis of hIAPP, unseeded or seeded by adding preformed fibrillar hIAPP, were compatible with models describing either fragmentation or secondary nucleation-dominated mechanisms (Supplementary Fig. 4). The latter mechanism has been highlighted to be important in the hIAPP fibril formation in vitro by several different groups previously[32,39–41]. In both cases, model analysis indicates that α-IAPP-O concentration-dependently delayed fibril formation by inhibiting primary nucleation, with minimal effects on fibril-dependent processes such as elongation, secondary nucleation, and fragmentation (Fig. 2d,e and Supplementary Figs. 5 and 6). This was further confirmed by experiments performed after pre-incubating hIAPP fibril seeds with α-IAPP-O (Fig. 2f). Consistent with selective inhibition of primary nucleation, a higher substoichiometric concentration of α-IAPP-O (10 hIAPP: 1 α-IAPP-O ratio) fully inhibited the formation of amyloid fibrils (Fig. 2g) by complexing prefibrillar oligomers with sizes ranging from 60 to 500 nm (Fig. 2h and Supplementary Fig. 7). Taken together, these data indicate that α-IAPP-O selectively targets hIAPP oligomers built at an early stage of the aggregation process (Supplementary Fig. 6e).

### α-IAPP-O prevents membrane damage of hIAPP oligomers and beta-cell apoptosis

Incubation of beta cells with hIAPP oligomers formed in vitro reduced the viability by 95% and increased apoptosis identified by TUNEL staining to 85% (Fig. 3a,b). Cytotoxicity was not observed when applying amyloid fibrils (Supplementary Fig. 8a,b) similar to previous findings[22,42]. α-IAPP-O neutralized cytotoxicity and apoptosis in a concentration-dependent manner as compared to an IgG control antibody. Beta cell apoptosis was accompanied by cell membrane deposition of hIAPP oligomers and ThioS-positive amyloid fibrils (Fig. 3c–e and Supplementary Fig. 8c–e). α-IAPP-O concentration-dependently inhibited the deposition of hIAPP on INS-1 beta cells (Fig. 3e and Supplementary Fig. 8e) and prevented permeabilization of liposome membranes induced by hIAPP oligomers (Fig. 3f, g). We next determined the effects of α-IAPP-O on human islets isolated from obese donors at risk for diabetes (Supplementary Table 2). Human islets were exposed to high glucose leading to the accumulation of extracellular ThioS-positive amyloid deposits and to beta cell apoptosis (Fig. 3h). Co-incubation with α-IAPP-O (0.5 μM) reduced ThioS-positive amyloid load and apoptotic beta cell death compared to IgG control. Further, α-IAPP-O improved beta cell function evaluated by insulin response to elevated glucose using islet perifusion, similar to

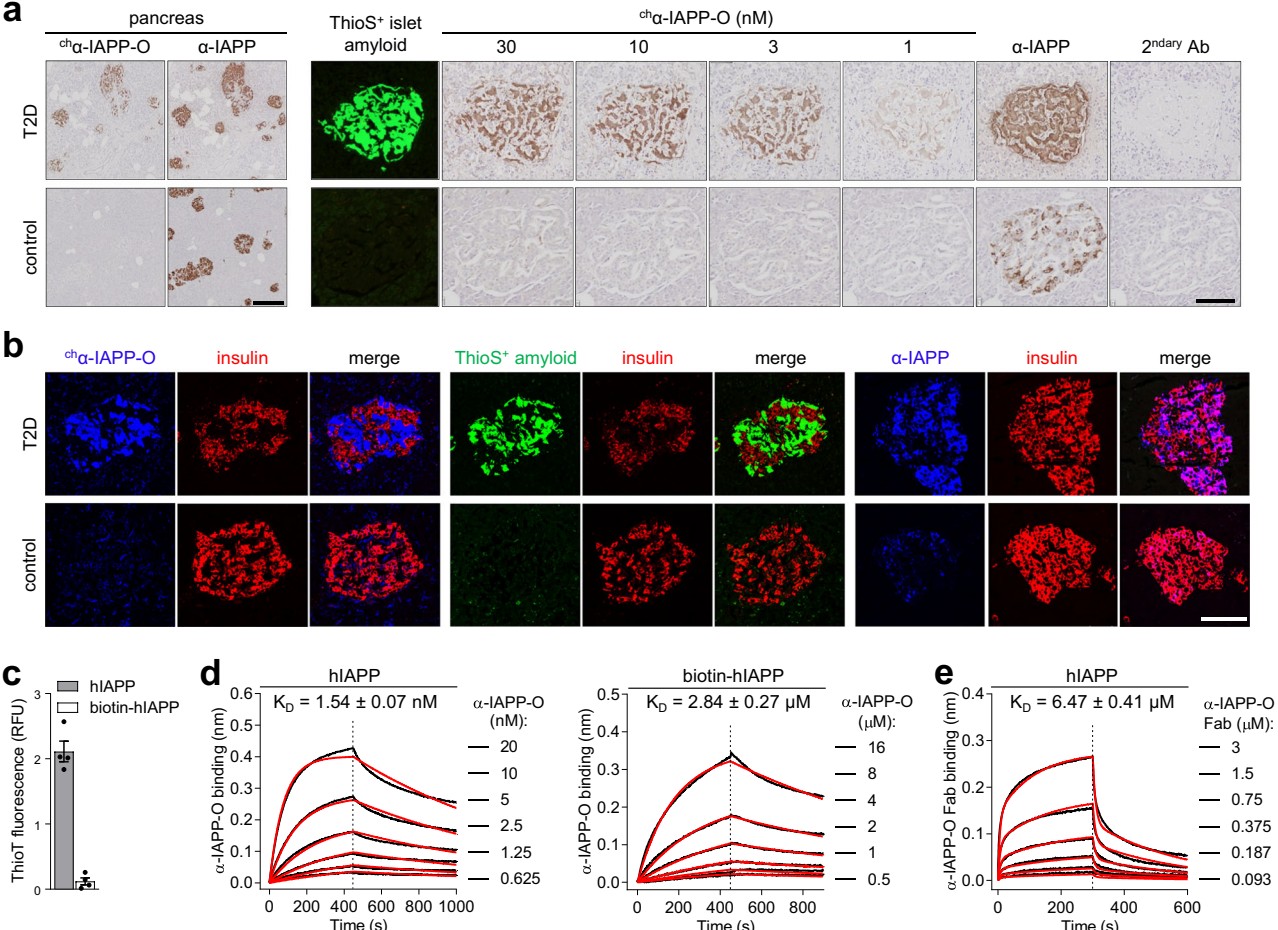

**Fig. 1 | α-IAPP-O antibody selectively binds to disease-relevant extracellular IAPP aggregates. a** Representative images of pancreas (left panel) and individual pancreatic islets (right panel) from a human type 2 diabetes (T2D) and a non-diabetic control subject stained with mouse chimeric α-IAPP-O (chα-IAPP-O, 30 nM or at indicated concentrations) and on adjacent sections with a mouse anti-IAPP antibody (α-IAPP, R10/99) that does not differentiate between monomeric and aggregated IAPP species (Supplementary Fig. 17). Islet amyloid fibrils were stained with thioflavin-S (ThioS) and anti-mouse secondary antibody was used as control (2ndary Ab). Scale bars: 300 and 100 μm, respectively. **b** Representative images of T2D and control human pancreatic islets stained with mouse chimeric α-IAPP-O (chα-IAPP-O, 30 nM; in blue, left panel), ThioS for extracellular amyloid deposits (in green, middle panel) and α-IAPP (E-5) for physiological human IAPP (in blue, right panel). Beta cells were visualized using anti-insulin antibody (in red) and merged

images are shown. Scale bar: 100 μm. Similar results for (**a**) and (**b**) have been obtained in at least three independent experiments on pancreatic tissue sections from different T2D and control subjects. **c** Aggregation propensity of human IAPP (hIAPP) and biotinylated human IAPP (biotin-hIAPP) in solution measured by thioflavin-T (ThioT) fluorescence. Data are means ± s.e.m. from four replicates. **d** Binding kinetic analysis of α-IAPP-O to hIAPP (left panel) and biotin-hIAPP (right panel) using biolayer interferometry (BLI). **e** BLI analysis of α-IAPP-O Fab binding to hIAPP. BLI dose-response association and dissociation curves (black line), fitting curves (red line) and $K_D$ constants (means ± s.e.m.) are derived from three independent runs (detailed in Supplementary Table 1). Association and dissociation are separated by a dashed line. Similar results for (**c**–**e**) have been obtained in three independent experiments.

the amyloid-inhibiting compound Congo red[43] at high concentration (25 μM) (Fig. 3i and Supplementary Fig. 8f).

## α-IAPP-O improves key pathological features of type 2 diabetes in rats and mice

The effects of α-IAPP-O were next evaluated in a transgenic rat model with beta cell-specific expression of hIAPP. Transgenic rats were shown to spontaneously develop a diabetic phenotype characterized by extensive islet amyloid formation resulting in progressive beta cell dysfunction and loss, ultimately leading to insulin depletion and hyperglycemia[15,16]. In this model, α-IAPP-O specifically engaged extracellular hIAPP oligomers surrounding insulin-producing beta cells in transgenic rat islets (Fig. 4a). The antibody did not bind to extracellular ThioS-positive amyloid, nor to monomeric IAPP constitutively expressed within transgenic and wild-type beta cells (Fig. 4a). Prediabetic transgenic rats characterized by emerging glucose intolerance (Supplementary Fig. 9a) were weekly administered with the rat

chimeric analog of α-IAPP-O (chα-IAPP-O, 3 mg/kg i.p.) demonstrating comparable binding to hIAPP and inhibition of amyloid formation (Supplementary Table 3 and Supplementary Fig. 10a). chα-IAPP-O reduced the progression of glucose intolerance and improved glucose-stimulated insulin secretion compared to vehicle after 8, 15, and 24 weeks of treatment with steady state plasma drug concentrations around 90 μg/ml (Fig. 4b, c and Supplementary Fig. 9a,b). The glucose-lowering drug metformin[44,45] (200 mg/kg/day in drinking water) improved glucose tolerance to a similar degree as did chα-IAPP-O but failed to stimulate insulin secretion upon oral glucose challenge. The combination of chα-IAPP-O and metformin had no cumulative effect on glucose control but ameliorated insulin secretion similarly to chα-IAPP-O monotherapy, indicating a direct protective effect of chα-IAPP-O on beta cell function (Supplementary Fig. 9c). Antibody treatment had no effect on islet size, beta cell area and islet amyloid deposition as compared to vehicle-treated glucose intolerant rats (Supplementary Fig. 9d). Moreover, chα-IAPP-O efficacy was target-related as it had no

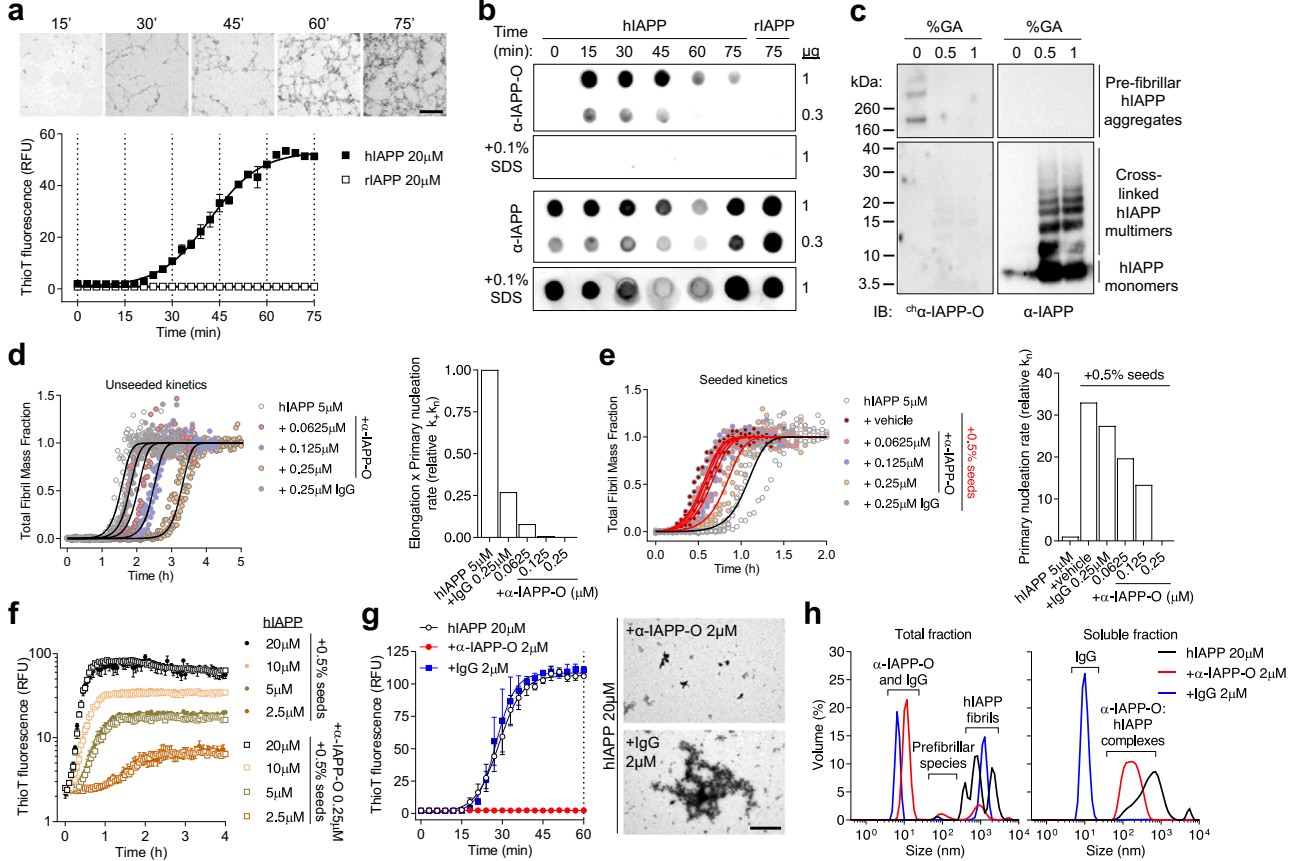

**Fig. 2 | α-IAPP-O interacts with transient pre-fibrillar hIAPP oligomers. a** Time course of amyloid fibril formation of hIAPP and rodent IAPP (rIAPP) in solution monitored by transmission electron microscopy (TEM; magnification: 1.05k; scale bar: 10 μm) and ThioT fluorescence. Data are means ± s.e.m. of five (hIAPP) and three (rIAPP) replicates. Dotted lines indicate time points of sampling for TEM and immunoblotting. **b** Dot blot analysis of fractions collected over the course of amyloid formation with α-IAPP-O and non-selective α-IAPP antibody (T-4145). (**c**) Immunoblotting of prefibrillar hIAPP aggregates and hIAPP monomers cross-linked with 0.5 and 1% glutaraldehyde (GA) using $^{ch}$α-IAPP-O and α-IAPP (T-4157) antibodies. **d** hIAPP aggregation kinetics in the presence of α-IAPP-O and IgG control antibody (left panel). Solid lines represent fitting to experimental data and the resulting product of elongation and primary nucleation ($k_+k_n$, right panel) using a model describing secondary nucleation-dominated aggregation. **e** Influence of α-IAPP-O on the aggregation kinetics of hIAPP seeded with 0.5% monomer equivalent preformed fibrils (red lines, left panel) and on the primary nucleation rate ($k_n$, right panel). Detailed kinetic modeling is shown in Supplementary Fig. 4, 5, and 6. **f** ThioT-monitored amyloid fibril formation as a function of initial hIAPP concentration supplemented with 0.5% seeds and 0.5% seeds pre-incubated with α-IAPP-O (0.25 μM). Data are means ± s.e.m. from triplicates. **g** Aggregation kinetics of hIAPP (20 μM) in the presence of α-IAPP-O and IgG control antibodies (2 μM) measured by thioflavin-T fluorescence (left panel). Data are means ± s.e.m of triplicates. Representative TEM images of hIAPP aggregates formed in the presence of α-IAPP-O and IgG control (right panel) after 60 min of incubation (dotted line on left panel). Scale bar: 2 μm. Similar data for (**a**–**g**) have been obtained in two (**b**–**g**) or three (**a**) independent experiments. **h** Size distribution in total and soluble fractions from the samples described in (**g**) (at 60 min) measured by dynamic light scattering (see also Supplementary Fig. 7). Data represent the average of three measurements performed on two biological replicates.

effect on glucose control nor body weight in wild-type rats. Applying an isotype control antibody had no impact on glycemia in transgenic or wild-type rats (Supplementary Fig. 9e/f). In a separate study, rats with marked glucose intolerance and hyperglycemia at baseline (Supplementary Fig. 11a,b) received weekly dosing of $^{ch}$α-IAPP-O (1, 3, and 10 mg/kg i.p.). $^{ch}$α-IAPP-O treatment was associated with significant improvements in glucose tolerance, glucose-stimulated insulin response and beta cell function at 1 and 10 mg/kg compared to vehicle (Fig. 4d–f and Supplementary Fig. 11c). These effects were accompanied by reduced glycemia, increased circulating insulin levels and normalized body weight (Fig. 4g–i and Supplementary Fig. 11d). $^{ch}$α-IAPP-O treatment was already effective after 7 weeks and effects became more apparent as the phenotype progressed (Supplementary Fig. 11e,f). Slowing of disease progression was associated with increased islet size and beta cell content in the pancreas of rats treated with $^{ch}$α-IAPP-O relative to vehicle (Fig. 4j,k and Supplementary Fig. 11g). $^{ch}$α-IAPP-O treatment was also associated with an increase in soluble IAPP levels and a decrease in insoluble IAPP aggregates in pancreas homogenates (Fig. 4l) that did not translate into reduced islet

amyloid deposits stained by thioflavin-S (Supplementary Fig. 11h). Further, we confirmed the therapeutic effects of α-IAPP-O in hIAPP transgenic mice characterized by glucose intolerance, hyperglycemia and insulin deficiency together with islet amyloidosis, oligomer deposition and beta cell loss (Supplementary Fig. 12). In this independent model, weekly administration of a mouse chimeric α-IAPP-O ($^{ch}$α-IAPP-O, 10 mg/kg i.p.) equivalent to the human IgG1 version (Supplementary Table 3 and Supplementary Fig. 10a), but not an isotype-matched control antibody improved glycemia and protected pancreatic beta cells (Supplementary Fig. 12b–h).

## α-IAPP-O removes extracellular hIAPP oligomers in transgenic rat islets

We further investigated the mechanism of action of α-IAPP-O in transgenic rats comparing $^{ch}$α-IAPP-O to an immunologically inert variant (inert $^{ch}$α-IAPP-O) carrying a mutated Fc region eliminating the interaction to rat Fc gamma receptors and silencing effector functions, while maintaining hIAPP binding properties (Supplementary Fig. 13) and inhibition potential of hIAPP aggregation

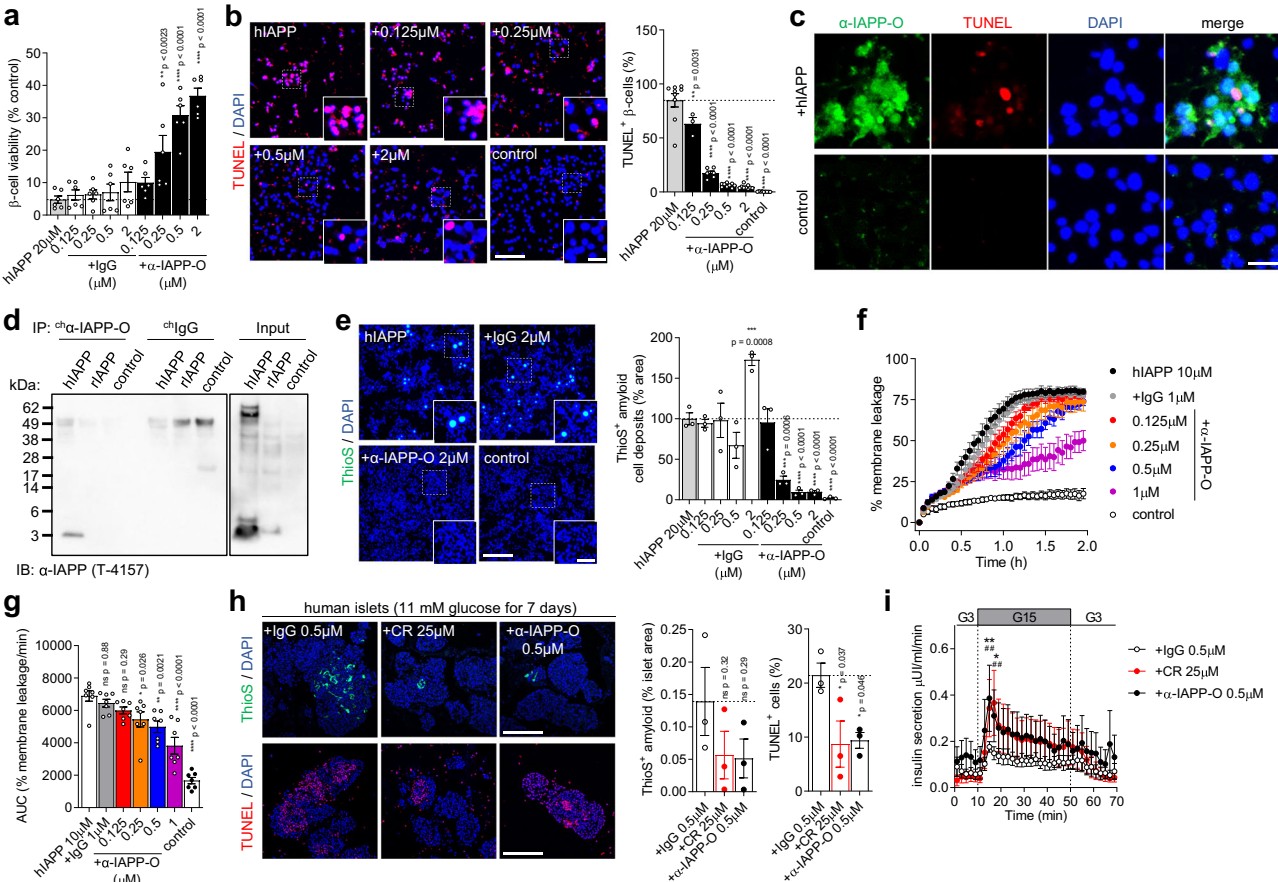

**Fig. 3 | α-IAPP-O protects beta cells from hIAPP oligomer-induced toxicity.**
**a** Viability of INS-1 cells exposed to hIAPP oligomers in presence of α-IAPP-O and IgG control. Data are from two independent experiments each performed in triplicates. **b** Representative images and quantification of hIAPP oligomer induced INS-1 cell apoptosis (red). Scale bars: 100 and 25 μm. Data are from three independent experiments (α-IAPP-O 0.125 μM $n = 1$ and for 0.25 μM $n = 2$ experiments) performed with three biological replicates. **c** Staining of membrane-bound hIAPP oligomers (green), and apoptotic INS-1 cell (red). Scale bar: 25 μm. For (**c, d**) similar results were obtained in an independent experiment. **d** Immunoprecipitation on INS-1 cell membrane extracts exposed to hIAPP oligomers, rIAPP, and control. **e** Representative images and quantification of amyloid (green) on INS-1 cells exposed to hIAPP oligomers supplemented with α-IAPP-O and IgG control. Scale bars: 300 and 100 μm. Data are from three biological replicates with similar data obtained in two independent experiments. hIAPP oligomer induced membrane

leakage in the presence of α-IAPP-O and IgG control over time (**f**) and resulting AUC (**g**). Data are from seven biological replicates from four independent experiments. **h** Representative images and quantification of ThioS-positive amyloid (green) and apoptosis (red) (scale bars: 50 μm) and (**i**) insulin secretion in high glucose treated human pancreatic islets in presence of α-IAPP-O, IgG control and Congo red (CR). Data in (**h,i**) are from three independent experiments, each using islets from a different donor (see Supplementary Table 2). For (**b, c, e, h**) DAPI stains nuclei (blue). Data are expressed as means ± s.e.m. Statistical analysis was done using one-way ANOVA and Dunnett's post hoc test for (**a, b, e, g, h**) with indicated $p$ values from multiple group comparison vs 20 μM hIAPP (**a, b, e**), 10 μM hIAPP (**g**) and vs IgG 0.5 μM (**h**), or repeated measures two-way ANOVA with Dunnett's post hoc test (**i**) with $**p = 0.0051$ and $*p = 0.0182$ (α-IAPP-O 0.5 μM versus +IgG 0.5 μM at 15 and 17 min), and $^{\#\#}p = 0.0040$ and $^{\#\#}p = 0.0037$ (CR 25 μM versus +IgG 0.5 μM at 15 and 17 min). ns not significant.

(Supplementary Fig. 10a, b). Pre-diabetic rats were dosed once a week with $^{ch}$α-IAPP-O (1, 3, and 10 mg/kg i.p.) and inert $^{ch}$α-IAPP-O (10 mg/kg i.p.). $^{ch}$α-IAPP-O dose-dependently reduced immunoreactive hIAPP oligomers by up to 68% relative to vehicle without significantly affecting ThioS-positive amyloid deposition in transgenic rat islets (Fig. 5a,b and Supplementary Fig. 14a,b). Likewise, $^{ch}$α-IAPP-O tended to decrease total pancreatic IAPP aggregates measured by ELISA (Supplementary Fig. 14c). Oligomer removal was paralleled by recruitment of CD68-positive islet resident macrophages to hIAPP oligomers and fibrils at the highest dose of $^{ch}$α-IAPP-O, without affecting the total number of islet macrophages (Fig. 5c and Supplementary Fig. 14d). In addition, clearance of hIAPP oligomers was associated with increased insulin-immunoreactive beta cell area and a trend for reduced levels of pro-inflammatory IL-1β in the pancreas (Fig. 5d and Supplementary Fig. 14e, f). In contrast, inert $^{ch}$α-IAPP-O did neither significantly affect hIAPP oligomer deposition, macrophage recruitment and IL-1β levels, nor preserved beta cell

content despite a small reduction in islet amyloid load (Fig. 5a–d and Supplementary Fig. 14a–f). Consistent with these findings and with the involvement of hIAPP oligomers in macrophage-mediated inflammation[27,46], α-IAPP-O dose-dependently stimulated Fc gamma receptor-mediated phagocytosis of hIAPP oligomers, and to a lesser extent of hIAPP fibrils, by human PBMC- and THP-1-derived macrophages in vitro, while decreasing macrophage IL-1β release (Fig. 5e and Supplementary Fig. 15).

## α-IAPP-O slows diabetes progression in human islet-engrafted mouse models

IAPP aggregation and amyloid deposition has been reported in transplanted human pancreatic islets where it might contribute to graft failure and recurrence of hyperglycemia[20,21,47]. To evaluate the effect of α-IAPP-O on graft function in vivo, immunodeficient NSG mice rendered diabetic by streptozotocin (STZ) injection or Rag2$^{-/-}$ mice fed a high-fat diet (HFD)[47,48] were both transplanted with human islets from

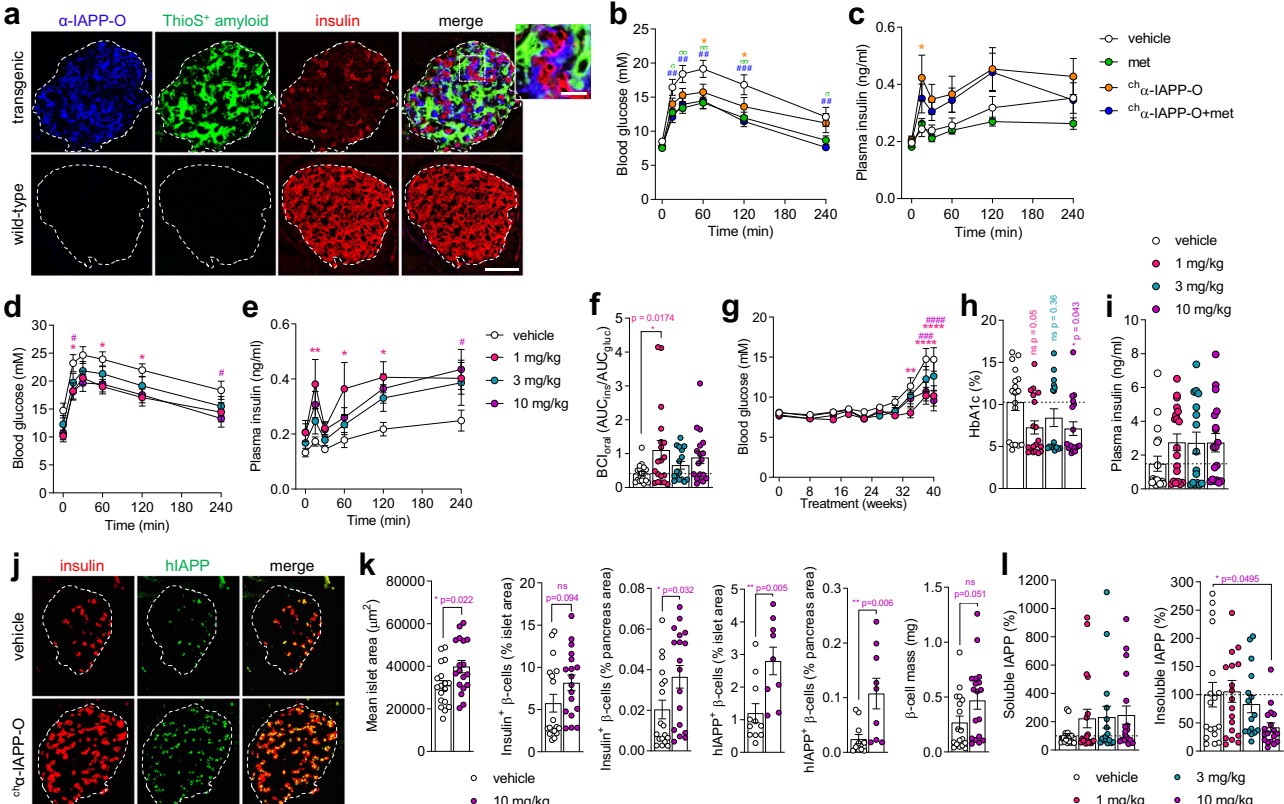

**Fig. 4 | α-IAPP-O delays diabetes progression and improves beta-cell function in hIAPP transgenic rats. a** Representative images of individual pancreatic islets from 57-week-old transgenic and wild-type rats stained for α-IAPP-O target engagement (blue), ThioS-positive islet amyloid (green) and insulin-positive beta cells (red). Scale bars: 100 and 25 μm. Similar results have been obtained in independent experiments using three different animals per genotype. **b** Blood glucose concentration and **c** plasma insulin response during oGTT in 36-week-old hIAPP transgenic rats injected with vehicle or rat chimeric ^chα-IAPP-O (^chα-IAPP-O, 3 mg/kg), metformin alone (met; 200 mg/kg/day) or in combination with 3 mg/kg ^chα-IAPP-O (^chα-IAPP-O+met) for 24 weeks. Group size for 4b-c were $n = 14$ vehicle, $n = 14$ met, $n = 16$ ^chα-IAPP-O, $n = 13$ ^chα-IAPP-O+met. **d** Blood glucose and **e** plasma insulin levels during oGTT in 50-week-old transgenic rats administered with ^chα-IAPP-O at 1, 3, and 10 mg/kg for 38 weeks compared to vehicle. **f** Beta cell function in rats after 38 weeks of treatment. **g** Longitudinal blood glucose levels, (**h**) HbA1c and (**l**) non-

fasting plasma insulin in transgenic rats after 41 weeks of treatment. Group size for (**d–i**), and (**l**) were $n = 18$ vehicle, $n = 19$ ^chα-IAPP-O 1 mg/kg, $n = 16$ ^chα-IAPP-O 3 mg/kg, $n = 18$ ^chα-IAPP-O 10 mg/kg. For (**f, g**) group averages from individual animals are shown. Representative images (**j**) and quantitative analysis (**k**) of pancreatic islets stained for insulin (red) and hIAPP (green; E-5 antibody) (scale bar: 100 μm), and (**l**) soluble IAPP and insoluble IAPP levels in pancreas homogenates after 41 weeks of treatment. Group sizes in (**k**) for vehicle and ^chα-IAPP-O 10 mg/kg are: 18/18 (mean islet area, insulin⁺ β-cells, β-cell mass) and 9/11 (hIAPP⁺ β-cells). For group sizes in (**l**) see above. All data are expressed as means ± s.e.m. * or ᵒ$p < 0.05$; ## or ᵒᵒ$p < 0.001$; ###$p < 0.0001$ (**b, c**) and * or #$p < 0.05$; **$p < 0.01$; ###$p < 0.001$; **** or ####$p < 0.0001$ by repeated measures ANOVA with Dunnett's test post-hoc (**b, c, d, e, g**), one-way ANOVA with Dunnett's post-hoc test (**f, h, i, l**), or two-tailed unpaired t test (**k**) comparing antibody/metformin-treated animals vs vehicle. ns not significant.

non-diabetic and pre-diabetic donors (Fig. 6a and Supplementary Table 2). While NSG recipient mice weekly administered with ^chIgG (10 mg/kg i.p.) rapidly returned to hyperglycemia, ^chα-IAPP-O (10 mg/kg i.p.) treatment maintained normoglycemia and delayed the recurrence of diabetes (Fig. 6b,c and Supplementary Fig. 16a). Human islet-engrafted Rag2⁻ᐟ⁻ mice fed a HFD 2 weeks post-transplant for 12 weeks developed glucose intolerance, hyperglycemia and hyperinsulinemia accompanying obesity, as opposed to non-obese recipients fed a control diet (Fig. 6d–g). Treatment with ^chα-IAPP-O (10 mg/kg i.p., once weekly) along with HFD normalized glycemia, and consistently reduced plasma insulin and human C-peptide levels (Fig. 6d–f and Supplementary Fig. 16b,c), pointing towards an improved function and adaptation of engrafted human islets. Furthermore, therapeutic treatment with ^chα-IAPP-O (10 mg/kg i.p., once weekly) initiated in obese diabetic mice previously fed a HFD for 6 weeks led to improved glucose tolerance (Fig. 6h) paralleled by stabilized fasting glucose levels (Fig. 6i) compared to recipients receiving ^chIgG (10 mg/kg i.p., once weekly). Graft analysis revealed a trend towards decreased oligomeric hIAPP but not amyloid deposits upon ^chα-IAPP-O treatment (Fig. 6j).

## Discussion

Clinically, the progressive nature of T2D is linked to beta cell dysfunction and loss, with patients lacking the ability to produce sufficient endogenous insulin to counteract insulin resistance and to control blood glucose levels[49]. While the primary cause of beta cell failure in T2D is unknown, the accumulation of aggregated forms of the beta cell peptide hormone IAPP in pancreatic islets is a likely contributor to decreased beta cell function and mass in an early stage of the disease. Here, we describe that hIAPP oligomers can cause beta cell dysfunction and death during T2D development which can be delayed by applying a human-derived antibody selective for these toxic hIAPP species in cultured beta cells and isolated human islets, as well as in transgenic rodents and human islet-engrafted mouse models.

The data prove that hIAPP prefibrillar oligomers applied to beta cells induce apoptosis in vitro, in contrast to monomers and mature fibrils. Neutralization of hIAPP oligomers by α-IAPP-O, a human monoclonal IgG1 antibody selectively binding extracellular hIAPP deposits in the pancreas of T2D patients and hIAPP intermediates produced during amyloid formation, prevented accumulation of amyloid fibrils, lipid membrane disruption and beta cell toxicity. This is

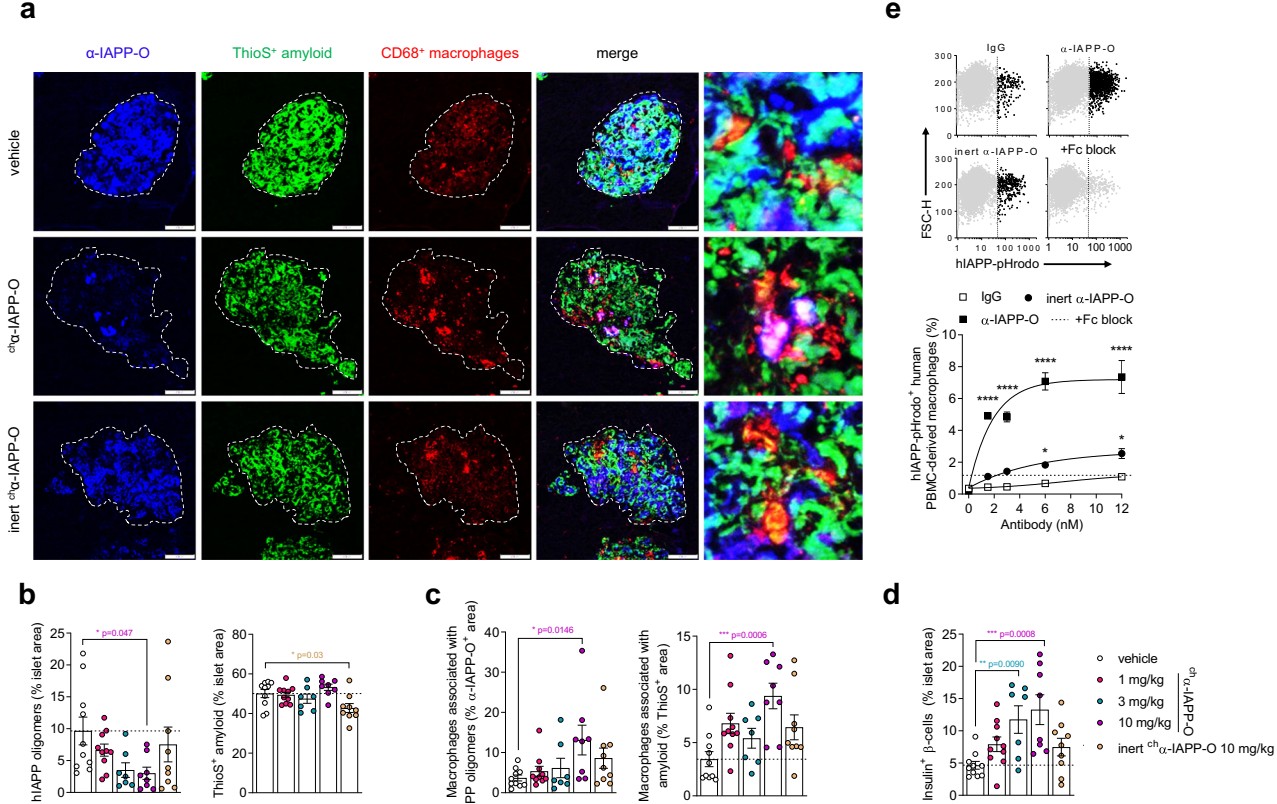

**Fig. 5 | α-IAPP-O stimulates macrophage-dependent clearance of hIAPP oligomers.** Representative images (**a**) and quantitative analysis (**b**–**d**) of islets from 50-week-old transgenic rats treated with $^{ch}$α-IAPP-O (1 mg/kg, 3 mg/kg, 10 mg/kg), inert $^{ch}$α-IAPP-O (10 mg/kg) and vehicle for 38 weeks. Pancreatic islets were stained for α-IAPP-O-positive oligomers (blue), ThioS-positive amyloid fibrils (green) and CD68-positive rat macrophages (red). Scale bars: 100 and 20 μm. **b** Quantification of hIAPP oligomer and amyloid fibrils in relation to islet area. **c** Quantification of CD68-immunoreactive macrophages associated with hIAPP oligomers and amyloid. **d** Quantitative analysis of insulin-immunoreactive beta cells in pancreatic islets. Final group numbers in (**a**–**d**) were $n = 10$ vehicle, $n = 10$ 1 mg/kg, $n = 7$ 3 mg/kg, $n = 8$ 10 mg/kg $^{ch}$α-IAPP-O, $n = 9$ inert $^{ch}$α-IAPP-O 10 mg/kg. **e** Uptake of hIAPP oligomers conjugated to the pH-sensitive dye pHrodo (hIAPP-pHrodo) by PBMC-derived human macrophages in the presence of α-IAPP-O, inert α-IAPP-O, and IgG control antibodies at increasing concentrations using flow cytometry. Blocking of Fc receptors expressed on macrophages was used to inhibit phagocytosis induced by α-IAPP-O at 12 nM (+Fc block, represented by a dashed line). Scatterplots of viable singlet macrophages plotted as a function of hIAPP-pHrodo versus forward scatter height (FSC-H) fluorescence intensities, and selection of hIAPP-pHrodo-negative (gray) or -positive (black) populations (left). Dose-response curves (right) showing increased phagocytosis of hIAPP oligomers by macrophages. Data are from three biological replicates. Similar data for (**e**) has been obtained in two independent experiments. All data are expressed as means ± s.e.m. Statistical analysis was done using one-way ANOVA followed by Dunnett's post hoc test for with indicated $p$ values derived from multiple group comparison vs vehicle (**b, c, d**), or ****$p < 0.0001$ by two-way ANOVA with Dunnett's test (**e**).

in line with previous studies indicating that toxic hIAPP oligomers deposit at the cell surface and permeabilize the cell membrane via pore formation and/or elongation into amyloid fibrils[50]. We have also shown that α-IAPP-O interacts with the N-terminus of IAPP when self-assembled into oligomers and inhibits the formation of amyloid fibril end-product, supporting a key role of IAPP N-terminal residues in the initial aggregation process facilitating membrane interaction and permeation[51]. Treatment with α-IAPP-O and with a general inhibitor of aggregation reduced islet amyloid content and beta cell toxicity in isolated human islets exposed to elevated glucose levels, strengthening the role of hIAPP oligomers and islet amyloidosis in human beta cell deterioration under diabetic conditions such as hyperglycemia. Of note, beta cell toxicity solely caused by hIAPP aggregates and independently of hyperglycemia was also reported in human and hIAPP-expressing mouse islets[52,53]. In these studies, beta cell apoptosis has been associated with oxidative stress, Fas upregulation and caspase-8 activation. Additional mechanisms by which hIAPP aggregates could mediate islet beta cell apoptosis and that are characteristic of human T2D include membrane disruption[54], endoplasmic reticulum (ER) stress[25], defects in autophagy[55], activation of the receptor for advanced glycation end-products (RAGE)[56] and inflammation[27,28,46]. These

pathways potentially contribute to hIAPP-induced beta cell loss and development of diabetes. This has been extensively studied in hIAPP-expressing transgenic rats and mice recapitulating features of human T2D[13–16]. In these animal models, both intracellular and extracellular hIAPP oligomers have been reported to trigger beta cell dysfunction and loss. Our data support a direct role for extracellular oligomers in beta cell pathogenesis, diabetes onset and progression in vivo. First, we have demonstrated that α-IAPP-O selectively engages extracellular oligomers in overtly diabetic hIAPP transgenic rats and mice after a single intraperitoneal injection. Oligomers bound by α-IAPP-O were extensively deposited around islet beta cells with a distribution distinct from amyloid fibrils. Second, chronic administration of α-IAPP-O in prediabetic and diabetic transgenic animals improved insulin secretion and glycemia accompanied with a trend towards increased beta cell mass. On that note, we cannot exclude that beta cell degranulation due to chronic hyperglycemia might contribute to the rather low insulin-positive area detected in transgenic animals.

α-IAPP-O also slowed beta cell failure and diabetes progression in human islet-engrafted mice, ruling out any confounding effects of hIAPP overexpression in transgenic models. While the mean data in the human islet engrafted models indicated delayed recurrence of

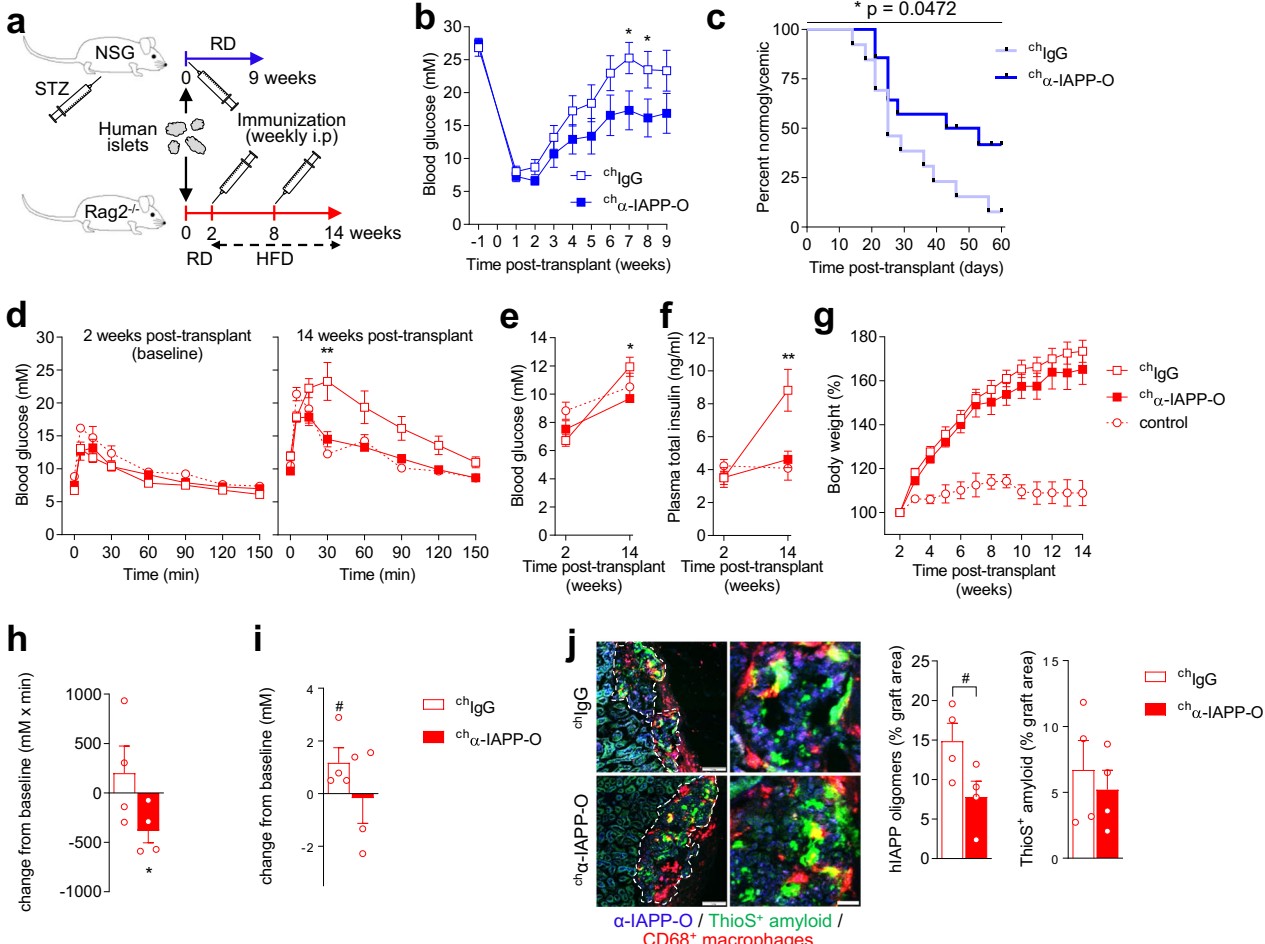

**Fig. 6 | α-IAPP-O improves human islet graft function in diabetic mice.**
**a** Experimental design to evaluate $^{ch}$α-IAPP-O pharmacological efficacy in human islet-engrafted NSG (**b**, **c**) and Rag2 null mouse models (**d**–**i**). Recurrence of hyperglycemia in NSG recipient mice treated with mouse chimeric α-IAPP-O ($^{ch}$α-IAPP-O, 10 mg/kg) and IgG control ($^{ch}$IgG, 10 mg/kg) measured as blood glucose levels (**b**) and percentage normoglycemic animals with blood glucose values < 13 mM (**c**). Data in (**b**, **c**) are from three independent experiments using human islets preparations from three non-diabetic donors (Supplementary Table 2) and final group numbers were $n = 13$ $^{ch}$IgG and $n = 14$ $^{ch}$α-IAPP-O. **d** Blood glucose concentration during oGTT, **e** fasting blood glucose and, **f** plasma insulin levels at baseline and 14 weeks post-transplant in HFD-fed Rag2$^{-/-}$ recipient mice treated with $^{ch}$α-IAPP-O (10 mg/kg) and $^{ch}$IgG control (10 mg/kg) for 12 weeks (14 weeks post-transplant). Naïve Rag2$^{-/-}$ recipients kept on regular diet (RD) served as controls. (**g**) Body weight of Rag2$^{-/-}$ recipient mice described in (**d**–**f**) expressed as percent increase from baseline. Data in (**d**–**g**) are from two independent

experiments using human islets preparations from pre-diabetic donors #5 and #6 (Supplementary Table 2) and final group numbers were $n = 9$ $^{ch}$IgG ($n = 8$ in **f**), $n = 6$ $^{ch}$α-IAPP-O and $n = 2$ control. **h** Change from baseline for glucose AUC from oGTT, (**i**) fasting glucose, and (**j**) immunofluorescence analysis of human islet grafts in HFD-induced diabetic Rag2$^{-/-}$ recipient mice treated with $^{ch}$α-IAPP-O (10 mg/kg, $n = 4$) and $^{ch}$IgG (10 mg/kg, $n = 4$) for 6 weeks (baseline at 8 weeks post-transplant). Human islets were isolated from a pre-diabetic donor #2 (Supplementary Table 2). (**j**) Representative images of human islet grafts from (**d**–**h**) and quantitative analysis stained for α-IAPP-O-positive oligomers (blue) and ThioS-positive amyloid (green). Scale bars: 100 and 20 μm. Data are expressed as means ± s.e.m. Statistical analysis was done using repeated measures ANOVA followed by Sidak's post hoc test (**b**, **d**, **h**), Log-rank Mantel-Cox test (**c**), two tailed unpaired t test (**e**, **f**, **h**), or one tailed Wilcoxon Signed Rank Test (**g**, **i**). *$p < 0.05$ (**b**, **c**, **e**, **h**); **$p < 0.01$ (**d**, **f**); #$p = 0.060$ (**i**) and #$p = 0.058$ (**j**). STZ streptozotocin, HFD high fat diet.

diabetes by α-IAPP-O, there was considerable variation among cohorts likely reflecting differences in human islet donors and preparations and proper adaptation of islets after transplantation. The therapeutic efficacy of α-IAPP-O might be limited by the inherent variability of this model with no or limited effects seen when islets fail to adapt after transplantation.

One mechanism of action of α-IAPP-O is believed to be mediated in part by the phagocytic clearance of extracellular hIAPP oligomers by islet macrophages in vivo. In addition, α-IAPP-O may also exert some of its effects by Fc-independent mechanisms via sequestering and neutralization of hIAPP oligomers, as observed with the inert version of the antibody lacking effector functions. Although α-IAPP-O promoted the recruitment of macrophages at sites of amyloid deposition, we did not observe any effect on amyloid load, consistent with the selective binding of α-IAPP-O toward hIAPP oligomers but not amyloid fibrils.

Other possible explanations are the limited capacity of macrophages to degrade and clear islet amyloid fibrils[57] and the constant buildup of amyloid fibrils in islets with a high functional beta cell mass[58].

α-IAPP-O and its chimeric derivatives inhibited ThioS-positive amyloid fibril formation in vitro and prevented amyloid deposition on cultured INS-1 beta cells and human islets. In transgenic mice and rats α-IAPP-O driven neutralization and macrophage-mediated clearance of toxic hIAPP oligomers was not associated with a clear effect on ThioS-positive IAPP amyloid. This might be due to the more complex environment in which hIAPP aggregation occurs in vivo, driven by a high overexpression of the transgene. Furthermore, differences in the experimental duration where amyloid formation is studied only for days in vitro and several months in vivo as well as differences in antibody exposure might be additional contributing factors. hIAPP oligomers are known inducers of NLRP3 inflammasome-mediated release of

IL-1β[28] and islet inflammation and islet macrophage activation are contributing factors to beta-cell dysfunction in T2D[27,44,57,59]. α-IAPP-O mediated neutralization of hIAPP oligomers led to a substantial reduction in IL-1β secretion in vitro and lowered IL-1β levels in pancreatic homogenates of treated animals. Overall, our data suggest that α-IAPP-O improves beta-cell function through neutralization and removal of toxic IAPP-oligomers as well as reduction of IAPP-mediated islet inflammation. Finally, α-IAPP-O might also exert a protective function on the islet vasculature as recent evidence suggests that hIAPP aggregation is a key factor contributing to cytotoxic and pro-inflammatory effects on endothelial cells and endothelial dysfunction in pancreatic islets[60].

We have demonstrated high selectivity of α-IAPP-O towards hIAPP oligomers with absent binding to physiological IAPP monomers and no effects on glucose control and body weight in wild-type rats. Still we cannot fully exclude potential non-islet effects of α-IAPP-O treatment linked to the presence of hIAPP deposits in extra-pancreatic organs such as kidney[61] and brain[62,63] which have been shown to alter microvasculature and mediate neuroinflammatory responses. Finally, the magnitude of the therapeutic effects of our approach might be underestimated due to the rapid disease progression and high overexpression of IAPP. Future studies in more physiological models such as miniature pigs[18,64] or non-human primates[65,66] could therefore be of additional translational value.

Together, our findings indicate that antibody-mediated removal of extracellular hIAPP oligomers in an early phase of events leading to beta cell exhaustion can preserve beta cell function and reduce the progression of T2D. This is of particular relevance since the effectiveness of currently available treatments for T2D is limited in time and likely impacted by the continuous deterioration of beta cell function over the course of the disease[67]. Alternative treatment strategies to delay disease progression by restoring and durably preserving beta cell function are needed[68,69]. Anti-diabetic medications offering blood glucose control by improving peripheral insulin sensitivity such as metformin, or by increasing insulin secretion such as dipeptidyl peptidase-4 inhibitors and glucagon-like peptide-1 receptor agonists (GLP-1 RAs) have demonstrated benefits in beta cell adaptation to high-fat diet-induced insulin resistance in hIAPP transgenic rodents and short-term improvements in human islet graft function in diabetic mouse recipients[44,70–72]. However, evidence for a long-term impact of these drugs on the progressive deterioration of beta cell function are lacking[73–76].

Initial attempts to generate pan-oligomer antibodies via active immunization with oligomers of the unrelated amyloid-β peptide (Aβ1−40) in hIAPP transgenic mice did not show beneficial effects on glucose control but rather increased β-cell apoptosis[77]. In contrast, active immunization approaches using stabilized IAPP oligomers improved glycemic control in hIAPP transgenic mice[78,79]. Moreover, passive immunization approaches in hIAPP transgenic mice using mouse-derived monoclonal antibodies against IAPP-protofibrils[80] or fibrils[81] delayed the progression of disease. Consistent with our findings, both studies reported increased insulin-positive area, and decreased β-cell apoptosis was found by Bortoletto and colleagues[80]. Overall, and in line with our data, IAPP-targeted antibody-based approaches seem to be safe although dedicated toxicology studies will be required before advancing such therapeutic approaches towards human testing.

Our approach of developing therapeutic antibodies is leveraging the immune system of elderly donors that stay healthy during the ageing process. Based on comprehensive analyses of their memory B-cell complements recombinant human antibodies are generated that are evolutionarily optimized and combine the advantages provided by human affinity maturation and tolerance selection. Using this approach, we were able to develop high-affinity antibodies with potent preclinical and clinical effects against a wide spectrum of misfolded and aggregated proteins such as beta-amyloid in Alzheimer's disease[82], alpha-synuclein in Parkinson's disease[30], SOD1 in amyotrophic lateral sclerosis[83], C9orf72 dipeptide repeat proteins in frontotemporal dementia[84], and transthyretin in transthyretin amyloid cardiomyopathy[85,86]. Our data with the human-derived antibody α-IAPP-O targeting toxic IAPP oligomers further expand this therapeutic concept towards T2D and opens a new avenue for beta cell protective therapies.

## Methods

The research performed complied with all relevant ethical regulations. Human blood samples from healthy elderly subjects were collected under written informed consent and study approval by the Ethics Committee of the Canton of Zürich. The study was compliant with the Helsinki Declaration. For human islet studies, donors or their families provided written informed consent and approval was obtained from the Human Research Ethics Board of the University of Alberta and the Institutional Ethical Committee of the University and the University Hospital of Lille. For animal experiments, handling, welfare, monitoring, and euthanasia practices were performed in strict accordance with the ethical guidelines. Animal studies were approved by the responsible authorities as specified below.

### Antibody generation

Antibodies were derived from a de-identified blood lymphocyte library collected from healthy elderly subjects by screening for high affinity binding toward aggregated hIAPP. Human memory B cells were isolated and cultured as previously described[82]. In short, memory B cells characterized by CD22+, CD27+, IgD−, IgM−, CD3−, CD56− and CD8a− markers were screened for the expression of antibodies binding to hIAPP aggregates and absence of cross-reactivity toward unrelated amyloid-forming proteins using ELISA. Selected hIAPP-reactive B-cell clones were subjected to cDNA cloning of IgG heavy and kappa or lambda light chain variable region sequences, and sub-cloned into human IgG1 constant domain expression vectors using Ig framework-specific primers for human variable heavy and light chain families in combination with human J-H segment-specific primers. Chimeric analogs (chα-IAPP-O) were engineered to contain mouse IgG2a or rat IgG2b backbones. Recombinant antibodies were transiently expressed in CHO-S cells and purified by protein A or protein G affinity chromatography. Fab fragments were generated by enzymatic digestion of human IgG1 antibody followed by purification on an IgG-CH1 affinity column (GingisKHAN Fab kit, Genovis).

### Preparation of IAPP peptides

Synthetic human IAPP (hIAPP), biotinylated hIAPP (biotin-hIAPP) and rodent IAPP (rIAPP) peptides (Bachem, Switzerland) were resuspended at 2 mg/mL in pure hexafluoro-isopropanol (HFIP, Sigma) with shaking overnight at room temperature, lyophilized using an Alpha 1−2 LDplus freeze dryer (Christ), and reconstituted in buffer to the desired concentration.

### Biolayer interferometry

Antibody binding kinetic was measured with an Octet RED96 instrument (Pall ForteBio). Lyophilized hIAPP was reconstituted in carbonate buffer (100 mM, pH 9.6) at 20 μg/mL or in acetate buffer (10 mM, pH 6) at 10 μg/mL and loaded on pre-equilibrated amino-propylsilane (APS) or activated amine-reactive (AR2G) biosensors (Pall ForteBio) according to manufacturer's recommendations. Biotin-hIAPP was reconstituted in kinetics buffer (Pall ForteBio) at 20 μg/mL and loaded on streptavidin (SA) biosensors. α-IAPP-O full IgG and Fab fragment were tested at indicated concentrations in PBS (pH 7.4) or kinetics buffer. Binding response relative to PBS, isotype control IgG, or Fab was analyzed with simultaneous Ka/Kd global fitting (2:1 or 1:1 interaction models) using the Octet system software.

BLI sensorgrams were drawn using GraphPad PRISM Version 9 (San Diego, USA).

## Aggregation assay and kinetic analysis

Spontaneous aggregation of hIAPP, biotin-hIAPP, and rIAPP peptides (Bachem) in the absence and presence of different concentrations of antibodies was assessed by monitoring amyloid fibril formation via the increase of fluorescence of the amyloid-specific dye thioflavin-T (ThioT, Sigma) over time. Lyophilized monomeric IAPP was reconstituted at 5 and 20 μM in serum-free RPMI 1640 medium (11879-RPMI, ThermoFisher Scientific). Reconstituted peptides were mixed with ThioT (5 mM in $H_2O$, filtered at 0.22 μm) to a final concentration of 20 μM in 96-well clear-bottom non-binding plates (Costar), and amyloid formation was recorded on a Varioskan LUX plate reader (ThermoFisher Scientific) measuring fluorescence emission at 489 nm (excitation at 456 nm; 12 nm bandwidth) every 3 min at room temperature (while shaking at 300 rpm for 10 s) or at a single time point corresponding to maximal fibril formation. Experimental data were analyzed and fractional fibrillar mass concentration and rate constants were calculated using the AmyloFit platform[87]. Microscopic events underlying hIAPP fibril formation that are inhibited by antibodies were identified by comparing rate constants obtained in the absence and presence of antibodies in unseeded and seeded reactions.

## Transmission electron microscopy

Samples were adsorbed onto glow-discharged carbon-coated copper grids (S162-3, Plano). Grids were stained with 2% (w/v) uranyl acetate for 1 min, washed with $ddH_2O$, air-dried, and imaged using a Philips CM100 transmission electron microscope with an acceleration voltage of 80 to 100 kV.

## Dot blot assay

Samples were diluted (1:3) in serum-free RPMI 1640 medium (11879-RPMI, ThermoFisher Scientific) ± 0.1% SDS and filtered through a nitrocellulose membrane (0.1 μm pore size). The membrane was washed in PBS, air-dried, blocked with PBS + 0.1% Tween® 20 (Sigma) + 5% BSA (Sigma) for 1 h, and incubated with α-IAPP-O (5 μg/ml) and a rabbit anti-IAPP primary antibody (α-IAPP, 2 μg/ml; T-4145, Peninsula Laboratories) in blocking buffer for 1 h at room temperature. After washing, the membrane was incubated with HRP-conjugated donkey anti-human and goat anti-rabbit secondary antibodies (#709-036-098 and #111-035-045, 1:10,000; Jackson ImmunoResearch), and revealed with chemiluminescent HRP substrate (Pierce™ ECL, ThermoFisher Scientific) using an ImageQuant LAS 4000 (GE Healthcare).

## Western blotting

Lyophilized hIAPP was reconstituted at 20 μM in serum-free RPMI 1640 medium (11879-RPMI, ThermoFisher Scientific) and mixed with glutaraldehyde (0, 0.5, or 1%; G6257, Sigma), incubated at 37 °C for 30 min with shaking, neutralized with 100 nM Tris-HCl pH 8 (2:1) and resolved under non-reducing and denaturing conditions by gradient SDS-PAGE (NuPAGE 4–12% Bis-Tris gels, Life Technologies) using NuPAGE LDS sample buffer (without reducing agent; Life Technologies) and NuPAGE MES running buffer (Life Technologies). Resolved proteins were electroblotted (Novex® Semi-Dry Blotter, 5 V, 1 h) onto PVDF membrane. Non-specific binding sites were blocked with PBS + 0.1% Tween® 20 (Sigma) + 2% BSA (Sigma) for 1 h. Membrane was immunoblotted with mouse chimeric $^{ch}$α-IAPP-O (10 μg/ml) and rabbit anti-IAPP antibody (α-IAPP, 1:500; T-4157, Peninsula Laboratories International) in blocking buffer for 1 h at room temperature, washed in PBS + 0.1% Tween® 20, and incubated with goat anti-mouse and goat anti-rabbit IgG (H + L) secondary antibodies coupled to HRP (#115-035-003 and #111-035-045, 1:10'000; Jackson ImmunoResearch). Antibody binding was revealed with HRP substrate (Pierce™ ECL, ThermoFisher

Scientific) and imaged using an ImageQuant LAS 4000 system (GE Healthcare, Switzerland).

## Dynamic light scattering

The average size of particles present in samples was measured by dynamic light scattering (DLS) at a fixed angle of $\theta = 173°$ and a laser source of 633 nm on a Zetasizer Nano (Malvern, UK). Additionally, samples were centrifuged at 10,000 g for 15 min and supernatants (soluble fractions) were analyzed. Samples were measured in triplicates in micro UV-Cuvettes with dimensions 12.5 × 12.5 × 45 mm (60 μl) and light path 1 cm (Brand GmbH, Germany).

## INS-1 cells and cell-based assays

Rat insulinoma INS-1 beta cells (INS-1 832/13, Cat# SCC207, Sigma) were grown in RPMI-1640 medium (30-2001, ATCC) supplemented with 10% fetal bovine serum (FBS), 100 μg/ml penicillin and 100 μg/ml streptomycin. Cultured cells were maintained in a humidified atmosphere of 5% $CO_2$ at 37 °C. Cells were seeded in 96-well plates (100 μl/well; ~60% confluency) 24 h prior incubation with hIAPP. Lyophilized hIAPP was reconstituted at 20 μM in serum-free RPMI 1640 medium (11879-RPMI, ThermoFisher Scientific) supplemented with 100 μg/ml penicillin and 100 μg/ml streptomycin. Human α-IAPP-O or IgG control antibody were added at indicated concentrations and peptide solutions were incubated for 4 h at room temperature under quiescent conditions before being applied to INS-1 β-cells. Unexposed cells were used as control.

Cell viability and apoptosis were assessed after 16 h. Cell viability was evaluated using MTT assay (MTT Cell growth Assay Kit, Merck Millipore) according to manufacturer's instructions with absorbance measured at 570 nm using a Varioskan LUX plate reader (ThermoFisher Scientific). Cell apoptosis was visualized by TUNEL staining (In Situ Cell Death Detection Kit, TMR red; Roche). Briefly, cells were fixed in 4% paraformaldehyde for 10 min at room temperature, washed in PBS, permeabilized in 0.25% Triton X-100 (in PBS) for 10 min and blocked with a solution containing 5% serum (horse/goat) + 4% BSA in PBS for 1 h, incubated with TUNEL solution (1:2 dilution; 50 μl/well) for 1 h at 37 °C, washed and kept in PBS for imaging. IAPP aggregates were stained on living cells exposed to human α-IAPP-O (200 nM) for 1 h prior to fixation and detected using Cy2-conjugated donkey anti-human secondary antibody (1:250, #709-225-149; Jackson ImmunoResearch).

Amyloid deposition was evaluated after 12 h by applying 0.01% Thioflavin-S (ThioS, Sigma) in $H_2O$ for 10 min at room temperature on fixed cells, with subsequent rinsing in ethanol 70% and $H_2O$. DAPI (1:1000) was included in the last washing step prior to imaging.

Fluorescence images were captured using a confocal laser scanning microscope (Leica SP8) and a widefield fluorescence imaging system (IN Cell Analyzer 2500 HS, GE Healthcare). Image analysis was performed on at least three different fields of view per well and three wells per condition using Image J software. Apoptosis was counted as the number of TUNEL and DAPI double-positive nuclei relative to total number of DAPI-positive nuclei and expressed as percentage. Amyloid deposition was computed as the image area occupied by ThioS staining expressed as percentage, with 100% corresponding to cells exposed to 20 μM hIAPP in the absence of antibody.

Membrane-bound IAPP aggregates were immunoprecipitated upon INS-1 cell exposure to hIAPP and rIAPP for 2 h. Unexposed cells served as control. Cells were rinsed in culture medium, gently scraped off the bottom of the wells and lysed by sonication. Cell lysates were incubated with mouse chimeric $^{ch}$α-IAPP-O (20 μg/ml) or $^{ch}$IgG control antibody (20 μg/ml) for 30 min at room temperature before addition of protein A-coated magnetic beads (Dynabeads, Life Technologies) for another 30 min at room temperature. After washing, antibody-antigen complexes were eluted from the beads in LDS sample buffer (Life Technologies) supplemented with 2.5% β-mercaptoethanol

(Sigma) by heating at 70 °C for 10 min and resolved by Western blotting.

## Membrane leakage assay

Liposomes (POPC/POPS at 7:3 molar ratio) encapsulated with 70 mM calcein disodium and formulated in 10 mM Tris−HCl, 100 mM NaCl (pH 7.4) were purchased from FormuMax Scientific Inc. (Sunnyvale, CA, USA). Leakage assay was performed in a 96-well clear bottom non-binding plate (Costar) in a total volume of 200 µL, consisting of 12 µL liposomes (0.4 mM), 8 µL of antibody in PBS for final concentrations indicated, and 180 µL hIAPP (10 µM) in 10 mM Tris−HCl, 100 mM NaCl. PBS was used as control in the absence of peptide and antibody. Calcein leakage was measured on a Varioskan LUX plate reader (ThermoFisher Scientific) by fluorescence emitted at 517 nm (495 nm excitation) every 3 min after plate was shaken for 9 s at 400 rpm. At the end of the experiment, maximum fluorescence leakage was induced by addition of 1 µL 10% Triton-X100 (Sigma). Percentage membrane leakage was calculated by the equation $(F - F_0)/(F_{max} - F_0) \times 100$, with $F$ corresponding to fluorescence measured over time, $F_0$ to initial fluorescence, and $F_{max}$ to maximum fluorescence.

## Human islet culture

Human pancreas tissue was harvested from obese adult brain-dead donors at the University Hospital Lille (France) and human islets were isolated as previously described[88]. In short, isolated human islets from different donors (Supplementary Table 2) were hand-picked and cultured in CMRL 1066 medium (Thermo Fisher Scientific) containing 5.6 mmol/l glucose supplemented with 100 U/mL Penicillin and 0.1 mg/mL streptomycin, 25 mmol/L HEPES (ThermoFisher Scientific), and 10% FBS for 6 days. Islets from donors # 1, #2, and #4 were further cultured in 11 mM glucose-containing CMRL media (supplemented with 100 U/mL penicillin and 100 µg/mL streptomycin) in the presence of human α-IAPP-O (0.5 µM), IgG control antibody (0.5 µM), or Congo Red (25 µM) for 7 days.

Islet function was evaluated using dynamic perifusion[89]. Briefly, 500 islets (500 IEQ in 500 µl) from donors #1, #2, and #3 were perifused (1 ml/min) with 3 mM glucose for 60 min (last 10 min used as baseline) and glucose-stimulated insulin secretion dynamics was determined at 15 mM glucose over 40 min. Insulin secretion was measured by chemiluminescent immunoassay (Access Ultrasensitive Insulin, Beckman Coulter) and calculated as insulin secretion (µUI/ml) per minute. Intra-islet insulin content is reported in Supplementary Fig. 8e.

Cultured islets were fixed in 4% (w/v) paraformaldehyde, paraffin-embedded, and cut in 3-µm sections for histological assessment of beta cell apoptosis and amyloid deposition. Sections were deparaffinized and rehydrated, permeabilized in 0.25% Triton X-100 (in PBS) for 10 min, blocked with a solution containing 5% serum (horse/goat) + 4% BSA in PBS for 1 h at room temperature, incubated with TUNEL solution (1:2 dilution; In Situ Cell Death Detection Kit, TMR red; Roche) for 1 h at 37 °C and with 0.15% (w/v) thioflavin-S (ThioS, Sigma) in $H_2O$ for 10 min, followed by rinsing in ethanol 70% and $H_2O$. DAPI (1:1000) was included in the last washing step prior slide mounting using Hydromount media (National Diagnostics). Slides were imaged on a Leica SP8 confocal laser scanning microscope. Image analysis was conducted on islets present on three sections (~50 µm interval) from each islet preparation using Image J software and the average of the three sections was calculated. The average islet count for the donor preparations #1, #2, and #4, and were: 58, 32, and 158, 158, 32, 58, for α-IAPP-O; 67, 16, and 37 for IgG control, and 56, 25, and 50 for congo-red (CR). Apoptosis was counted as the number of TUNEL and DAPI double-positive nuclei relative to total number of DAPI-positive nuclei. Amyloid deposition was computed as the islet area occupied by ThioS staining.

## Transgenic rat studies

Hemizygous hIAPP transgenic male rats (RIP-HAT; CD:SD-Tg(ins2-IAPP)Soel) and wild-type male Sprague-Dawley rats (SD) were obtained from Charles River Laboratories (Germany) and housed under controlled conditions (22 ± 2 °C, 12:12 h light/dark cycle with light phase from 2:00 a.m. to 2:00 p.m.; 40–60% humidity) with free access to standard chow diet (Extrudate 3436, KLIBA NAFAG, Switzerland) and water. Rats were randomized based on body weight and blood glucose concentration during oral glucose tolerance test and received a once-weekly intraperitoneal (i.p.) injection of recombinant rat chimeric antibody (chα-IAPP-O) or PBS at a volumetric dose of 2 ml/kg. Rats were weighed every week to determine the dose of antibody injected. Metformin (1,1-Dimethylbiguanide hydrochloride, 97%; D150959, Sigma) was supplied in drinking water (3–3.8 g/L). Daily water intake was estimated by weighing the water bottles and metformin concentration was adjusted accordingly to reach a target dose of 200 mg/kg/day. Treatments started at 12 weeks of age and were blinded until full completion of the studies.

Oral glucose tolerance test (oGTT) was performed on fasted rats (12 h overnight fasting with free access to water). Rats were orally administered with 2 g/kg glucose (50% solution, B. Braun Medical AG) and blood samples were repeatedly collected (0, 15, 30, 60, 120, and 240 min) from the sublingual vein under gas anesthesia (3% isoflurane, Zoetis, Switzerland). Blood glucose and glycated hemoglobin (HbA1c) were measured using a Contour XT glucometer (Bayer) and A1CNow+ test kit (Bayer), respectively. Plasma was isolated by centrifugation and insulin levels were determined by ELISA (rat insulin ELISA, Mercodia). Beta cell function was estimated by $BCI_{oral} = AUC_{insulin}/AUC_{glucose}$ (AUC, area under the concentration curves during oGTT)[90].

Following a washout period of 3 weeks without any treatment (equivalent to the plasma half-life of rat chα-IAPP-O), and 3 days prior to sacrifice, rats received a single intraperitoneal administration of human α-IAPP-O (30 mg/kg, i.p.) for histological measurement of α-IAPP-O-bound hIAPP aggregates in pancreatic islets (in vivo target engagement). Rats were euthanized by sodium pentobarbital injection (60 mg/kg, i.p.) and pancreas tissue was removed, weighted, and cut in three parts corresponding to pancreas head, core and tail. A small piece of pancreas tail was dissected out, weighted, flash frozen in liquid nitrogen, homogenized (10%, w/v) in modified RIPA buffer (50 mM Tris-HCl pH 7.4, 150 mM NaCl, 1 mM EDTA, 5% NP-40, 0.5% Sodium deoxycholate, 0.1% SDS) and centrifuged (100,000 g, 4 °C, 30 min). Supernatant was collected and pellet was solubilized by sonication in 70% formic acid. Supernatants and pellets were kept stored at −80 °C until analyzed for soluble and insoluble IAPP content by ELISA (EZHA-52K, Millipore). The other parts of the pancreas were fixed in 4% paraformaldehyde (w/v) and embedded in paraffin, or frozen in OCT medium following soaking in 30% sucrose for histological analyses. Animal experiments were approved by the Veterinary Office of the Canton Zurich, Switzerland (authorization number 143/2015) and performed as recommended by the Federal Veterinary Office (FVO).

## Human islet transplant into NSG and Rag2$^{-/-}$ mice

NSG male mice (NOD.Cg-*Prkdc$^{scid}$ Il2rg$^{tm1Wjl}$*/Sz, #005557, The Jackson Laboratory, USA) aged 6–20 weeks, and made diabetic by intraperitoneal injection of 180 mg/kg streptozotocin, were transplanted with 280-340 human islets under the kidney capsule. Human islets from non-diabetic, cadaveric donors (Supplementary Table 2) were obtained from Prodo Laboratories (Aliso Viejo, USA) and the Alberta Diabetes Institute IsletCore (Edmonton, Canada), hand-picked to 90–95% purity and cultured overnight with 10 µg/mL mouse chα-IAPP-O or isotype control antibody in complete CMRL media (containing 100 U/mL penicillin, 100 µg/mL streptomycin, 0.05 mg/mL gentamicin, and 2 mmol/L glutamax) prior to transplantation. Animals were maintained under controlled conditions (14:10 h light/dark cycle, at 22 ± 1 °C and 50–60% humidity) on chow diet (6% fat, Teklab 2918, Huntingdon, UK).

Rag2 null male mice (B6.129S6-*Rag2$^{tm1Fwa}$* N12, #RAGN12-M, Taconic Biosciences, Lille Skensved, DK) were maintained under controlled conditions (12:12 h light/dark cycle, 20–24 °C and 40–60% humidity) on chow diet (LFD, NJ D12450B, Research Diets, New Brunswick). At 13–57 weeks of age, mice were transplanted with 450 human islets isolated from non-diabetic brain-dead donors (obtained from the University Hospital Lille, France; additional information in Supplementary Table 2) and placed on a high-fat diet (HFD, NJ D12450B, Research Diets, New Brunswick) 2 weeks post-transplant.

Mice were injected with 10 mg/kg mouse $^{ch}\alpha$-IAPP-O or isotype control antibody intraperitoneally (i.p.), starting 1 day prior transplant (NSG mice) and 2- or 8-weeks post-transplant (Rag2 null mice), and subsequently once weekly for the duration of the studies. Oral glucose tolerance test (3 g/kg glucose) and blood glucose measurement were performed following a 5-h fast. Plasma total insulin (mouse and human) and human C-peptide levels were determined by ELISA (mouse insulin and human C-peptide ELISA, Mercodia). Mice were injected with human $\alpha$-IAPP-O (30 mg/kg, i.p.) 3 days before sacrifice to measure the amount of hIAPP aggregates in human islet grafts. For NGS-mice, transplant experiments were terminated at 9 weeks post-transplant unless recipient mice reached humane endpoint based on body weight loss and prolonged hyperglycemia. Human islet-engrafted recipient mice were euthanized by cervical dislocation following deep anesthesia. Graft-bearing kidneys were frozen in OCT medium, and 5-μm sections were cut for histology. Studies were approved by the Animal Care Committee of the University of British Columbia, and by the institutional ethical committee of the University of Lille.

## Histology

Formalin-fixed paraffin-embedded human (obtained from University Hospital Basel, Switzerland) and rat pancreas, and OCT-embedded fresh rat pancreas and human islet graft-bearing kidneys were cut in 5 μm-sections. Paraffin-embedded sections were deparaffinized, rehydrated, and immersed in antigen retrieval solution (70% formic acid) for 10 min. For DAB-based staining, endogenous peroxidase activity was quenched with 3% $H_2O_2$ in methanol for 10 min. Fixed and frozen sections were blocked in PBS + 2.5% horse serum + 2.5% goat serum + 4% BSA for 1 h, and incubated with mouse chimeric $^{ch}\alpha$-IAPP-O, mouse monoclonal anti-IAPP (1:100; R10/99, Abcam), mouse monoclonal anti-human IAPP (1:100; E-5, Santa Cruz Biotechnology), guinea pig polyclonal anti-insulin antibody (1:3; FLEX, Dako), mouse monoclonal anti-rat CD68 (1:1000; MCA341GA, Bio-Rad), and rat monoclonal anti-mouse CD68 antibody (1:200; ab53444, Abcam). Peroxidase-based staining was performed using biotinylated donkey anti-mouse secondary antibody (1:500; Jackson ImmunoResearch) combined with Vectastain ABC detection (Vector Laboratories). Fluorescence detection was achieved using Cy5- (#715-175-150, 1:200), Cy3- (#715-165-150, 1:200), TRITC-conjugated goat anti-guinea pig (#106-025-003, 1:200), and Cy3-conjugated mouse anti-rat (#212-165-082, 1:200) secondary antibodies (all from Jackson ImmunoResearch). Amyloid deposits were stained using 0.15% (w/v) thioflavin-S (ThioS, Sigma) in $H_2O$ followed by rinsing in ethanol 70% and $H_2O$. $\alpha$-IAPP-O-bound hIAPP aggregates (in vivo target engagement) were revealed on fresh frozen tissue sections using Cy5-conjugated donkey anti-human secondary antibody (#709-175-149,1:200, Jackson ImmunoResearch). Slides were mounted using Hydromount media (National Diagnostics). Bright-field imaging was performed on a Dotslide VS120 slide scanner (Olympus) and fluorescence imaging was performed on a Leica SP8 confocal laser scanning microscope. Islet area, insulin- and hIAPP-immunoreactive beta cell content were quantified on paraffin-embedded rat pancreas sections. For each animal, at least 25 islets >2500 μm$^2$ in size were analyzed on four sections each from pancreas head, core, and tail region. $\alpha$-IAPP-O-bound hIAPP aggregates, ThioS-positive amyloid, and CD68-immunoreactive macrophages present within islets were

quantified on rat pancreas and human islet graft cryosections (four and three to six sections per tissue, respectively). Data were computed as the fluorescence area above a predetermined threshold using Image-Pro Premier software (Media Cybernetics) and expressed as percentage of the corresponding islet and tissue area. Colocalization between CD68-immunoreactive macrophages and $\alpha$-IAPP-O-bound hIAPP aggregates or ThioS-positive amyloid was analyzed using ImageJ/FiJi software. Beta cell mass (mg) was calculated as follows: (Σinsulin-positive area/pancreas area) × pancreas weight (mg).

## Phagocytosis assay

Peripheral blood mononuclear cells (PBMCs) were enriched from healthy donor blood using monocyte isolation kit (Miltenyi Biotec) and differentiated into macrophages in serum-free medium (M-SFM, ThermoFisher Scientific) supplemented with 100 ng/mL GM-CSF (Gibco) and 100 μg/ml penicillin/streptomycin for 6 days at 37 °C and 5% $CO_2$. PBMC-derived macrophages were plated ($5 \times 10^5$ cells/well) and cultured in M-SFM medium supplemented with 100 μg/ml penicillin/streptomycin, 100 ng/mL GM-CSF, 1 ng/mL LPS (Sigma) and 20 ng/mL IFN-γ 1 day prior to the experiment.

Lyophilized hIAPP peptide was reconstituted in 0.1 M sodium bicarbonate buffer (pH 8.4) to a final peptide concentration of 4 mg/ml, incubated with pHrodo green STP ester dye (20 mg/ml in DMSO; P35369, ThermoFisher Scientific) for 30 min at room temperature in the dark, lyophilized with an Alpha 1-2 LDplus freeze dryer (Christ) and stored at −20 °C until use. Lyophilized pHrodo-labeled hIAPP and unlabeled hIAPP were reconstituted (1:3) in serum-free RPMI-1640 medium (Gibco) to a final concentration of 5 and 15 μM, respectively. Human $\alpha$-IAPP-O, inert $\alpha$-IAPP-O or IgG control antibody were added at various concentrations, followed by incubation for 2 h at room temperature under quiescent conditions.

PBMC-derived macrophages were incubated with a 1:10 dilution of the solution containing pHrodo-labeled hIAPP aggregates (20 μM) and antibodies (0, 1.5, 3, 6, and 12 nM) in fresh M-SFM medium supplemented with 100 μg/ml penicillin/streptomycin and 50 μg/mL of the scavenger receptor inhibitor Fucoidan (F5631, Sigma) for 30 min at 37 °C. Human Fc receptor (FcR) blocking solution (1:10 dilution; 130-059-901, Miltenyi Biotec) and cytochalasin D (50 μg/ml; C2618, Sigma) were added to inhibit FcR-dependent and general phagocytosis. After detachment, macrophages were washed in PBS and fluorescence of any surface-bound pHrodo-labeled hIAPP was quenched by addition of trypan blue (10%), and phagocytosis was analyzed using a FACS Aria II flow cytometer equipped with BD FACS Diva software (BD Biosciences). Intracellular pHrodo green was excited using a 488 nm laser and the fluorescence emission was collected using a 530/30 nm filter (FITC). A minimum of 10,000 events were acquired from each sample and data were exported as Flow Cytometry Standard format 3.0 files (FCS files) and analyzed with FlowJo software (Tree Star Inc.). Gating was done on single macrophages with high forward and side scatter (FSC-A and SSC-A) levels, and pHrodo-hIAPP-positive macrophages with fluorescence emission above cytochalasin D-treated macrophages (negative control) were counted.

## Statistical analysis

Data are expressed as means ± s.e.m and results between groups were analyzed using two-tailed Student's t-test, one-way and two-way ANOVA with post hoc tests for multiple comparisons if not disclosed otherwise. Statistical analyses were conducted using GraphPad PRISM 9 (GraphPad Software, USA) and significance was set at *$p < 0.05$, **$p < 0.01$, ***$p < 0.001$, and ****$p < 0.0001$.

## Reporting summary

Further information on research design is available in the Nature Portfolio Reporting Summary linked to this article.

## Data availability

The main data supporting the results of this study are available within the paper and its supplementary information. Source data are provided with this paper.

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

## Acknowledgements

We are grateful to Winnie Gut, Kerstin Linder, Leoni Hugentobler, Erika Tarasco, Ines Rito Brandao, Alexandra Durrer, Maik Krüger, Marie Kopp, Fulvio Grigolato, Anna Jeske, Daniel Schuppli, Petra Allenspach, Luzia Senn, and Nadine Glassl for technical help, Benoît Combaluzier for help on antibody discovery and production processes, Fabian Buller for reviewing the manuscript and Tobias Welt for conceptual contribution. We thank the Human Organ Procurement and Exchange (HOPE) program, BC Transplant, and Trillium Gift of Life Network (TGLN) for their work in procuring human donor pancreas for research, and the Alberta Diabetes Institute IsletCore for their efforts in human islet isolation. We especially thank the organ donors and their families for their kind gift in support of diabetes research. We acknowledge staff members of the Center for Microscopy and Image Analysis of the University of Zurich (ZMB). This work was supported by CTI grants 13750.1 PFFLE-LS and 17448.1 PFLS-LS to M.Y.D., T.A.L., and R.M.N, by Canadian Institutes of Health Research grants PJT-165943 and PJT-156449 to C.B.V., and by JDRF postdoctoral fellowships 3-PDF-2014-191-A-N and 3-APF-2018-586-A-N to H.C.D.

## Author contributions

F.W. and F.D.H. conceived experiments and analyzed the data. F.W. and F.D.H. prepared the manuscript. F.W., F.D.H., C.S., I.C., and K.B. performed experiments. H.C.D and J.T. conducted the human islet graft experiments. H.C.D., M.O., P.A., J.K-C., F.P., C.B.V., T.A.L. contributed to the experimental design and interpretation of the results. M.Y.D. provided human tissues and critically reviewed the manuscript. C.H., R.M.N., and J.G. supervised the project. All authors approved the final version of the manuscript.

## Competing interests

F.W., F.D.H., C.S., I.C., K.B., C.H., R.M.N. and J.G. are or were (F.D.H. and C.S.) employees and shareholders of Neurimmune; F.W., F.D.H., I.C., and J.G. are inventors on patents related to this work. All other authors do not declare any conflict of interest.
