## [Peer Review File · Nature Communications]

A human antibody against pathologic IAPP aggregates protects beta cells in type 2 diabetes modelsEditorial Note: This manuscript has been previously reviewed at another journal that is not operating a transparent peer review scheme. This document only contains reviewer comments and rebuttal letters for versions considered at *Nature Communications*.

REVIEWER COMMENTS

Reviewer #1 (Remarks to the Author):

In this revision Wirth et al present data on a novel human monoclonal antibody that targets and neutralizes amyloid forming oligomers of human IAPP. They show that it can block the aggregation of these oligomers and so prevent the toxic effects on beta cells in vitro. They also show longterm weekly IP injections of the antibody into transgenic rats and mice expressing the hIAPP dampened the progression to impaired glucose tolerance. They transplanted human islets in immune compromised mice (STZ diabetic NSG and HFD Rag2 null) to translate their finding to human tissue. They suggest that antibody mediated removal of IAPP oligomers could be a pharmacological strategy for treating type 2 diabetes.

It appears that they have added new data, but there is no clear indication that is so. There has been some effort to address my raised issues but unfortunately not all adequately addressed. Restated they are:

1. The lack of evidence of protected beta cell mass.

The phrases “Beta cell protection”, “prevent beta cell loss” are used throughout:

“Slowing of disease progression was associated with preserved islet size and beta cell content (Fig 4j,k)” (line 293+); In Discussion (line 484+) chronic treatment “in prediabetic and diabetic transgenic animals reduced beta cell loss”; in hIAPP transgenic mice “improved glycemia and protected pancreatic beta cells(Fig S12b-h)”(lines 299+).Fig 4 title.

However, the data for prevention of loss are still weak. While there are modest benefits in glucose tolerances compared to vehicle / control, there few instances of normalized blood glucose, just improved. In Fig 4 (rats)there is increased mean islet area but neither the insulin positive beta cells as % of islet area or the beta cell mass differed between the anti-IAPP ab and iverhicle control, even using a large sample number (n=22-25 mice per group). In Fig S12 for the transgenic mice the mean islet area is increased but again insulin + beta cells

as % of islet area or % of pancreatic area or beta cell mass are significantly different than either vehicle or IgG control. Additionally in both the % of insulin + beta cells/ islet is extremely low in rats 5-8% with maximum 16% and mouse mean of 15 with maximum of 40-50%; one expects in adult rodents about 75-85% of the islet area is beta cell. This discrepancy may be due to degranulation from chronic hyperglycemia or some technical issue but is concerning.

2. As I mentioned before, "the number of islets/ cells counted are not included, just "all the islet on 3 sections 50 um apart", which would be recounting many of the same islets since most rodent are 50-175 diameter. They show much larger islets based on their magnification bars so it would be likely to recount. They now have written a more detail but still the same issue. There is only comment re how close the sections are for the cultured human islets, (line 722+) "image analysis was conducted on all islets present on 3 sections (~50 um interval) from each islet preparation". For the others:

In Methods, (line 818+) for the rat studies "islet area, insulin and hIAPP immunoreactive beta cell content were analyzed on all islets (> 2500um²) identified ... pancreas head, core and tail (4 sections each per rat)" (That minimal area is about 50-60 um diameter islet.) On line methods (399+)" Islet area and insulin immunoreactive beta cell content were quantified from all islets identified on 5 sections per mouse" .

3. I believe the human islet studies are new but there are issues here too. When islets from non diabetic humans were transplanted (Fig 6b and FigS16a) in STZ diabetic NSG mice , the anti IAPP ab treatment " maintained normoglycemia and delayed the recurrence of diabetes" (Lines 394+). However while Fig6b showed data only to 7 weeks, the more complete data in FigS16a goes to 9 wks and shows the results from the individual 3 donors: the first has delayed hyperglycemia becoming hyperglycemic (above 10mM) by 9 weeks, the second shows no difference between the antibody and the control isotype, and the third with 280 islets shows no difference but with 340 islets is not significantly different but shows a trend (very large SEMs for n=4 recipient mice). Thus the data actually do not show consistently delayed recurrence of diabetes. In HFD fed Rag 2 null mice transplanted with prediabetic human islets, the antibody is said to "reverse the abnormal glucose intolerance"(Line 402+). Fig 6h has only one time point with significant difference and the

control data only given in FigS16d looks superimposable on the anti-IAPP treated data. Why are these data not presented together in Fig 6?

4. In Fig3A and B there is discrepancy in viability. In A with 2um anti-IAPP ab treatment the beta cell viability is only 40% of that of control but in the images, treatment with the ab has almost not TUNEL positivity.

In Figure 3H, one wonders if much of the TUNEL positivity is from central necrosis of the islets in the very large islets shown after 7 days in 11 mM glucose. The quantification shown with 8-10% TUNEL + cells in treated and 20% in control (IgG) treated would suggest that the images are not representative.

In Fig 3I the perfusion data for human islets. It is not usual to present data for insulin secretion in perfusion as % from baseline. One might expect different baselines from the different conditions; it would be more informative to show it as insulin/ml/min.

Minor:

Figure 5 Panel B is duplicated over part of the images of panel a.

Throughout the figure legends there are (F(1,13)=8.321) notations without any explanation these are distracting.

Reviewer #2 (Remarks to the Author):

The authors were very responsive to the original review and have extensively revised the manuscript as a result. The additional data and clarity of the response is really appreciated and the revisions to the text of the manuscript are thoughtful and improve the manuscript.

There remain still some parts that are not fully addressed - the data supporting cell membrane localization could still be strengthened, and the lack of availability of some of the requested data is unfortunate.

However, taken together, these remaining issues are relatively minor and do not detract from what is an impressive body of work which provides novel, impactful data for the field.

Reviewer #3 (Remarks to the Author):

The manuscript characterizes the efficacy of an antibody which bind specifically to oligomeric human IAPP. The concept is very interesting and have potential translation. The same concept has been published by Bortoletto et al in Adv Sci 2022 showing that mAb can delay disease progression and increases overall survival and 2017 by Bram et al at Sci Rep. This concept is controversial as other reports such as Lin Cy in Diabetes showed that vaccination induced high titers of hIAPP antibodies but b cell apoptosis was increased. Antibodies were isolated from human B cells from elderly donors. The rational of using healthy elderly donors is not explained. For example, why not using T2D donors. If these complexes are present in T2D I would assume antibody response against them and possibly higher affinity candidates. They cloned the IgG genes and expressed them transiently. I am not clear why the antibody fragment, Fab was not expressed recombinantly but rather isolated from IgG1 by enzymatic digestion rather than expressed as recombinant protein. Antibody was shown block the aggregation and toxicity of beta cells in vitro. The in vivo experiments showed that antibody reduced progression of glucose intolerance. They hypothesize removal of IAPP oligomers by the antibody could be a pharmacological strategy for treating type 2 diabetes. The in vivo data however demonstrate modest effect. So still long way to go yet. I also wonder why type 1 diabetes was not tested as well. I would assume hIAPP would be present in islet of type 1 diabetes.

Comments

The figure legend describing the in vivo experiments are massive and is difficult to follow. It will be good to shorten the legend, so it is easier to follow.

Point-by-point response on revised NCOMMS-23-03297-T

Manuscript: “A human antibody against pathologic IAPP aggregates protects beta cells in models of type 2 diabetes”

We thank the reviewers for their critical review of our manuscript and their valuable suggestions. We have addressed all comments from all reviewers and revised the manuscript accordingly. Please find our point-to-point response below.

Reviewers' expertise:

Reviewer #1: islet biology, diabetes

Reviewer #2: islet biology, IAPP

Reviewer #3: expertise in antibody engineering

Reviewers' comments:

Reviewer #1 (Remarks to the Author):

In this revision Wirth et al present data on a novel human monoclonal antibody that targets and neutralizes amyloid forming oligomers of human IAPP. They show that it can block the aggregation of these oligomers and so prevent the toxic effects on beta cells in vitro. They also show longterm weekly IP injections of the antibody into transgenic rats and mice expressing the hIAPP dampened the progression to impaired glucose tolerance. They transplanted human islets in immune compromised mice (STZ diabetic NSG and HFD Rag2 null) to translate their finding to human tissue. They suggest that antibody mediated removal of IAPP oligomers could be a pharmacological strategy for treating type 2 diabetes.

It appears that they have added new data, but there is no clear indication that is so. There has been some effort to address my raised issues but unfortunately not all adequately addressed. Restated they are:

1. The lack of evidence of protected beta cell mass.

The phrases “Beta cell protection”, “prevent beta cell loss” are used throughout: “Slowing of disease progression was associated with preserved islet size and beta cell content (Fig 4j,k)” (line 293+); In Discussion (line 484+) chronic treatment “in prediabetic and diabetic transgenic animals reduced beta cell loss”; in hIAPP transgenic mice “improved glycemia and protected pancreatic beta cells (Fig S12b-h)”(lines 299+). Fig 4 title.However, the data for prevention of loss are still weak. While there are modest benefits in glucose tolerances compared to vehicle / control, there few instances of normalized blood glucose, just improved. In Fig 4 (rats)there is increased mean islet area but neither the insulin positive beta cells as % of islet area or the beta cell mass differed between the anti-IAPP ab and iverhicle control, even using a large sample number (n=22-25 mice per group). In Fig S12 for the transgenic mice the mean islet area is increased but again insulin + beta cells as % of islet area or % of pancreatic area or beta cell mass are significantly different than either vehicle or IgG control.

Response: We agree with reviewer 1 that the emphasis on beta cell death/loss might be overstated. We believe that the improvement in glycemia following antibody treatment seen throughout the different in vivo studies likely reflects both amelioration of IAPP-induced beta cell dysfunction as well as attenuating beta cell loss.

We have modified the manuscript accordingly:

Line 288-290, page 14 (Result Section)

«Slowing of disease progression was associated with ~~preserved~~ increased islet size and beta cell content in the pancreas of rats treated with ^{ch}α-IAPP-O relative to vehicle.»

Line 301 , page 15 (Title Figure 4, Result Section)

«Figure 4 α-IAPP-O delays diabetes progression and ~~beta-cell loss~~ improves beta cell function in hIAPP transgenic rats.»

Line 471-472 and line 477-478, page 23 (Discussion Section)

“Second, chronic administration of α-IAPP-O in prediabetic and diabetic transgenic animals ~~reduced beta-cell loss and~~ improved insulin secretion and glycemia accompanied with a trend towards increased beta cell mass (formerly “to increase beta cell mass”). [...]”

Line 475-476, page 23 (Discussion Section)

“α-IAPP-O also ~~prevented~~ slowed beta cell failure and diabetes progression in human islet-engrafted mice, ruling out any confounding effects of hIAPP expression in transgenic models.»

Line 502-504, page 24 (Discussion Section, additional sentence added)

«Overall, our data suggest that α-IAPP-O’s improves beta cell function through neutralization and removal of toxic IAPP-oligomers as well as reduction of IAPP-mediated islet inflammation.»

Additionally in both the % of insulin + beta cells/ islet is extremely low in rats 5-8% with maximum 16% and mouse mean of 15 with maximum of 40-50%; one expects in adult rodents about 75-85% of the islet area is beta cell. This discrepancy may be due to degranulation from chronic hyperglycemia or some technical issue but is concerning.

Response: As described in methods, the percentage insulin-positive area was deduced by quantifying the fluorescence area above a predetermined threshold using Image-Pro Premier automated image analysis. A fixed threshold was predetermined to obtain an optimal signal to noise ratio irrespective of the genotype and treatment and was applied throughout the entire analysis. To exclude potential technical artefacts, we have quantified the insulin-positive area for an independent set of wild-type and transgenic rats using the identical settings. This analysis revealed an insulin-positive area of 75% in the expected range of wild-type rats, and 8-12% in transgenic rats, ruling out a technical artefact of the quantification (please find a set of representative images below). The target engagement study depicted in Figure S12a was performed in young cohort of in hIAPP transgenic mice with an early diabetic phenotype and a corresponding higher insulin-positive area.

We agree with the reviewer that we cannot exclude that the low insulin-positive area in the transgenic models could in part result from a degranulation due to chronic hyperglycemia. We have added an additional sentence in the discussion to state this possibility:

Line 472 – 474, page 23 (Discussion Section)

“On that note, we cannot exclude that beta cell degranulation due to chronic hyperglycemia might contribute to the rather low insulin-positive area detected in transgenic animals.”

2. As I mentioned before, “the number of islets/ cells counted are not included, just “all the islet on 3 sections 50 μ m apart”, which would be recounting many of the same islets since most rodent are 50-175 diameter. They show much larger islets based on their magnification bars so it would be likely to recount. They now have written a more detail but still the same issue. There is only comment re how close the sections are for the cultured human islets, (line 722+) “image analysis was conducted on all islets present on 3 sections (~50 μ m interval) from each islet preparation”. For the others: In Methods, (line 818+) for the rat studies “islet area, insulin and hIAPP immunoreactive beta cell content were analyzed on all islets (> 2500 μ m²) identified ... pancreas head, core and tail (4 sections each per rat)” (That minimal area is about 50-60 μ m diameter islet.) On line methods (399+) “ Islet area and insulin immunoreactive beta cell content were quantified from all islets identified on 5 sections per mouse” .

Response: We have updated the methods section accordingly to include this additional information:

Line 732-736, page 33-34 (Method Section)

“Image analysis was conducted on islets on three sections (50 μ m interval) from each islet preparation using Image J software and the average of the three sections was calculated. The average islet count for the donor preparations H1045, H1065, and H1073 were: 158, 32, 58, and for α -IAPP-O; 37, 16, and 67 for IgG control, and 50, 25, and 56 for congo-red (CR).”

Line 833-836, page 37 (Method Section)

“Islet area, insulin- and hIAPP-immunoreactive beta cell content were quantified on paraffin-embedded rat pancreas sections. For each animal, at least 25 islets $> 2500 \mu\text{m}^2$ in size were analysed on 4 sections each from pancreas head, core and tail region.”

Line 390-392, page 29 (Method Section, Supplementary Material)

“Islet area and insulin-immunoreactive beta cell content were quantified from at least 25 islets identified per section on a total of five sections per mouse.”

3. I believe the human islet studies are new but there are issues here too. When islets from non diabetic humans were transplanted (Fig 6b and FigS16a) in STZ diabetic NSG mice , the anti IAPP ab treatment “ maintained normoglycemia and delayed the recurrence of diabetes” (Lines 394+). However while Fig6b showed data only to 7 weeks, the more complete data in FigS16a goes to 9 wks and shows the results from the individual 3 donors: the first has delayed hyperglycemia becoming hyperglycemic (above 10mM) by 9 weeks, the second shows no difference between the antibody and the control isotype, and the third with 280 islets shows no difference but with 340 islets is not significantly different but shows a trend (very large SEMs for n=4 recipient mice). Thus the data actually do not show consistently delayed recurrence of diabetes.

Response: The reviewer is correct on the different observation periods for the individual engrafted cohorts using the STZ diabetic NSG mice. While mice engrafted with human islet from R140, HP-15315-UI and HP-16111-01 (340 islets) were followed up to 9 weeks, the cohort engrafted with 280 islets of HP-16111-01 could only be monitored up to 7 weeks. This cohort was terminated at 7 weeks due to the severe diabetic stage of the control group, reaching humane endpoint criteria (loss of body weight and elevated hyperglycemia in untreated animals). We have updated Figure 6b to now shows blood glucose levels up to 9 weeks post-transplant for the other cohorts. Furthermore, we have updated the text as follows:

Line 803-805, page 36 (Method section):

“For NGS-mice, transplant experiments were terminated at 9 weeks post-transplant unless mice reached humane end point based on body weight loss and prolonged hyperglycemia.»

Regarding the consistency of effects of α -IAPP-O to delay re-occurrence of diabetes in the NGS-mice we deliberately showed the individual human donor cohorts to show the high degree of variability between donors. It is challenging to obtain an ideal “sub-optimal” transplant with the inherent donor/preparation variability. The mean data clearly showed delayed recurrence, but we wanted to be transparent and show the donor-to-donor and experiment-to-experiment variability in this challenging model. We have added a corresponding sentence to the discussion:

Line 477-479, page 23 (Discussion section):

“While the mean data in the human islet engrafted models clearly showed delayed recurrence of diabetes by α -IAPP-O, there was considerable variation among cohorts likely reflecting differences in human islet donors and preparations. »

In HFD fed Rag 2 null mice transplanted with prediabetic human islets, the antibody is said to “reverse the abnormal glucose intolerance”(Line 402+). Fig 6h has only one time point with significant difference and the control data only given in FigS16d looks superimposable on the anti-IAPP treated data. Why are these data not presented together in Fig 6?

Response: To better illustrate α -IAPP-O's therapeutic effect we have updated Figure 16 and now show change from baseline for glucose tolerance (Figure 16h) and fasting glucose levels (Figure 16i) for both, control and α -IAPP-O's treated animals.

Accordingly, we have updated the corresponding text in the result section now reading as follows:

Line 390-393, page 19 (Result Section)

"Furthermore, therapeutic treatment with ^{ch} α -IAPP-O (10 mg/kg i.p., once weekly) initiated in obese diabetic mice previously fed a HFD for six weeks ~~reversed~~ led to improved glucose tolerance (Fig. 6h) paralleled by stabilized fasting glucose levels (Fig. 6i) compared to recipients receiving ^{ch}IgG (10 mg/kg i.p., once weekly)."

Third, the number of islets/ cells counted are not included, just "all the islet on 3 sections 50 um apart", which would be recounting many of the same islets since most are 50-175 diameter.

Response: Please see our comments and revisions above.

4. In Fig3A and B there is discrepancy in viability. In A with 2um anti-IAPP ab treatment the beta cell viability is only 40% of that of control but in the images, treatment with the ab has almost not TUNEL positivity.

Response: The reviewer correctly states that the antibody effects on cell viability measured via MTT are less pronounced than that effects on apoptosis assessed via TUNEL. This might be explained by the fact that opposed to TUNEL assessing cell death, MTT is measuring cell metabolism and cell number, which reflects on both cell death and proliferation. Hence, effect sizes of IAPP on these two different readouts are likely to be different (see also Potter et al., PNAS, 2010). Consequently, different therapeutic effects sizes are likely. The antibody could be protecting almost fully from IAPP-induced beta cell death (TUNEL), but only partly preventing the decreased proliferation and function that is induced by the high concentration of hIAPP.

In Figure 3H, one wonders if much of the TUNEL positivity is from central necrosis of the islets in the very large islets shown after 7 days in 11 mM glucose. The quantification shown with 8-10% TUNEL + cells in treated and 20% in control (IgG) treated would suggest that the images are not representative.

Response: As we did not observe TUNEL positivity in isolated human islets from three independent donors cultured under normoglycemic conditions we attribute the observed TUNEL staining to the prolonged high glucose environment and increased IAPP secretion rather than central necrosis. We agree with the reviewer on the selection of images and have included more representative examples in Figure 3h.

In Fig 3I the perfusion data for human islets. It is not usual to present data for insulin secretion in perfusion as % from baseline. One might expect different baselines from the different conditions; it would be more informative to show it as insulin/ml/min.

Response: We have changed the presentation of data in Figure 3i to insulin/ml/min.

Minor:

Figure 5 Panel B is duplicated over part of the images of panel a. Throughout the figure legends there are $(F(1,13)=8.321)$ notations without any explanation these are distracting.

Response: We thank the reviewer for the input on the figure duplication which has been fixed. Furthermore, we have updated all figure legends omitting all F-value and t-values originating from ANOVA and t-tests and only reporting the p-values from direct comparison or post-hoc analysis for better readability.

Reviewer #2 (Remarks to the Author):

The authors were very responsive to the original review and have extensively revised the manuscript as a result. The additional data and clarity of the response is really appreciated and the revisions to the text of the manuscript are thoughtful and improve the manuscript.

There remain still some parts that are not fully addressed - the data supporting cell membrane localization could still be strengthened, and the lack of availability of some of the requested data is unfortunate.

However, taken together, these remaining issues are relatively minor and do not detract from what is an impressive body of work which provides novel, impactful data for the field.

Response: We thank reviewer 2 for the kind words and the time and efforts invested to revise and improve our manuscript.

Reviewer #3 (Remarks to the Author):

The manuscript characterizes the efficacy of an antibody which bind specifically to oligomeric human IAPP. The concept is very interesting and have potential translation. The same concept has been published by Bortoletto et al in Adv Sci 2022 showing that mAb can delay disease progression and increases overall survival and 2017 by Bram et al at Sci Rep. This concept is controversial as other reports such as Lin Cy in Diabetes showed that vaccination induced high titers of hIAPP antibodies but b cell apoptosis was increased.

Response: We thank the reviewer for pointing out these publications which we now discuss in the revised manuscript:

Line 531-541, page 25 (Discussion Section)

“Initial attempts to generate pan-oligomer antibodies via active immunization with oligomers of the unrelated amyloid- β peptide (A β P1–40) in hIAPP transgenic mice did not show beneficial effects on glucose control but rather increased β -cell apoptosis⁷⁹. In contrast, active immunization approaches using stabilized IAPP oligomers improved glycemic control in hIAPP transgenic mice^{80,81}. Moreover, passive immunization approaches in hIAPP transgenic mice using mouse-derived monoclonal antibodies against IAPP-protofibrils⁸² or fibrils⁸³ delayed the progression of disease. Consistent with our findings, both studies reported increased insulin-positive area and decreased β -cell apoptosis was found by Bortoletto and colleagues⁸². Overall, and in line with our data, IAPP-targeted antibody-based approaches seem to be safe although dedicated toxicology studies will be required before advancing such therapeutic approaches towards human testing.”

Antibodies were isolated from human B cells from elderly donors. The rationale of using healthy elderly donors is not explained. For example, why not using T2D donors. If these complexes are present in T2D I would assume antibody response against them and possibly higher affinity candidates.

Response: The reviewer raises an important point. Our strategy of developing therapeutic antibodies for protein aggregation diseases is based on the analyses of memory B-cell repertoires of healthy elderly donors. Based on this approach we have discovered a large spectrum of pharmacologically active human-derived antibody molecules against aggregation-prone proteins such as beta-amyloid, alpha-synuclein, SOD1, C9orf72 dipeptide repeat proteins and transthyretin (references 30 and 84-88). As spontaneous protein misfolding and aggregation events are expected to occur to a baseline extent also during healthy ageing, we are aiming at leveraging such potentially protective antibodies. We agree with the reviewer that also in overt type 2 diabetes, corresponding antibodies are likely to exist, and it would be interesting to study such antibody complements in patients in future studies.

We have amended the discussion to better explain the rationale of using healthy donors:

Line 542-553, page 25 (Discussion Section)

“Our approach of developing therapeutic antibodies is leveraging the immune system of elderly donors that stay healthy during the ageing process. Based on comprehensive analyses of their memory B-cell complements recombinant human antibodies are generated that are evolutionarily optimized and combine the advantages provided by human affinity maturation and tolerance selection. Using this approach, we were able to develop high affinity antibodies with potent preclinical and clinical effects against a wide spectrum of misfolded and

aggregated proteins such as beta-amyloid in Alzheimer's disease⁸⁴, alpha-synuclein in Parkinson's disease³⁰, SOD1 in amyotrophic lateral sclerosis⁸⁵, C9orf72 dipeptide repeats in frontotemporal dementia⁸⁶, and transthyretin in transthyretin amyloid cardiomyopathy^{87,88}. Our data with the human-derived antibody α -IAPP-O targeting toxic IAPP oligomers further expand this therapeutic concept towards T2D and opens a new avenue for beta cell protective therapies."

They cloned the IgG genes and expressed them transiently. I am not clear why the antibody fragment, Fab was not expressed recombinantly but rather isolated from IgG1 by enzymatic digestion rather than expressed as recombinant protein.

Response: The reviewer correctly states that Fabs can be produced by recombinant expression as alternative to proteolytic digestion. As only small quantities of Fab were required for these experiments and we have optimized the recombinant expression and purification of large quantities of full-length IgG, we applied the approach of enzymatic digest which results in the rapid generation of fragments with equivalent binding properties and purity.

Antibody was shown block the aggregation and toxicity of beta cells in vitro. The in vivo experiments showed that antibody reduced progression of glucose intolerance. They hypothesize removal of IAPP oligomers by the antibody could be a pharmacological strategy for treating type 2 diabetes. The in vivo data however demonstrate modest effect. So still long way to go yet. I also wonder why type 1 diabetes was not tested as well. I would assume hIAPP would be present in islet of type 1 diabetes.

Response: The reviewer raises an important point that aggregated IAPP could be a relevant target also for the treatment of type 1 diabetes. While the role of IAPP oligomers and fibrils in type 2 diabetes is well studied, less is known of about their contribution to human type 1 diabetes. Several studies detected IAPP amyloid in pancreatic biopsies of patients with type 1 diabetes (Beery et al., Islets, 2019; Westermark et al., Ups J Med Sci, 2017). Moreover, formation of islet amyloid was observed in islets transplanted into patients with type 1 diabetes potentially contributing to islet inflammation, dysfunction and graft failure (Denroche and Verchere, J Mol Endocrin, 2018). While this was outside of the scope of the current manuscript, we agree that future studies are warranted to address this in more detail.

Comments

The figure legend describing the in vivo experiments are massive and is difficult to follow. It will be good to shorten the legend, so it is easier to follow.

Response: We have shortened the figure legends for better readability.

REVIEWERS' COMMENTS

Reviewer #1 (Remarks to the Author):

In this second revision of Wirth et al, the authors are now more precise in what they show, albeit in a somewhat misleading way. They revised it so there is a vague “protection” of beta cells. They do show their antibody treatment results in improved glycemia, increased % insulin area/pancreas (a relative value, dependent on area/volume/mass of pancreas) and mean islet area but in no case do they show changes in % insulin area /islet area or .most importantly, changes in beta cell mass.

They still overstate the findings with transplanted human islets, stating (line 477) “ while mean data ...clearly showed delayed recurrence of diabetes...., there were considerable variation among cohorts likely reflecting differences in human islet donors and preparations”. Again, as shown in Suppl Fig 16a, there are only 2 time points with significance in only 1 of the 3 donors. The cohort from the first has delayed hyperglycemia becoming hyperglycemic (above 10mM) by 9 weeks, the second shows no difference between the antibody and the control isotype, and the third with 280 islets shows no difference but with 340 islets is not significantly different but shows a trend. Thus to say there is a clear delay is not accurate.

Line 54 also needs to be rewritten since the wording is incorrect and not what was in the cited review or the references cited there. Amyloid deposits are seen in the majority of patients with T2D, not in “the majority of islets from patients with T2D” as stated.

Reviewer #3 (Remarks to the Author):

Fantastic work, really enjoy reading it. I have no specific comment, but a question: Did you try doing some structural studies? Also, can authors give more details on how they selected the antibodies?

Point-by-point response on final revision of NCOMMS-23-03297B

Manuscript: “A human antibody against pathologic IAPP aggregates protects beta cells in type 2 diabetes models”

We thank the reviewers for their further critical review of our revised manuscript and their further suggestions. We have addressed the remaining comments and revised the manuscript accordingly. Please find our point-to-point response below.

Reviewers' expertise:

Reviewer #1: islet biology, diabetes

Reviewer #3: expertise in antibody engineering

Reviewers' comments:

Reviewer #1 (Remarks to the Author):

In this second revision of Wirth et al, the authors are now more precise in what they show, albeit in a somewhat misleading way. They revised it so there is a vague “protection” of beta cells. They do show their antibody treatment results in improved glycemia, increased % insulin area/pancreas (a relative value, dependent on area/volume/mass of pancreas) and mean islet area but in no case do they show changes in % insulin area /islet area or, most importantly, changes in beta cell mass.

They still overstate the findings with transplanted human islets, stating (line 477) “ while mean data ...clearly showed delayed recurrence of diabetes...., there were considerable variation among cohorts likely reflecting differences in human islet donors and preparations”. Again, as shown in Suppl Fig 16a, there are only 2 time points with significance in only 1 of the 3 donors. The cohort from the first has delayed hyperglycemia becoming hyperglycemic (above 10mM) by 9 weeks, the second shows no difference between the antibody and the control isotype, and the third with 280 islets shows no difference but with 340 islets is not significantly different but shows a trend. Thus to say there is a clear delay is not accurate.

Response: We thank the reviewer for his additional comments. We are confident that the totality of data presented based on multiple independent and complementary *in vitro* and *in vivo* models provides a compelling body of evidence for the therapeutic effectiveness of antibody α -IAPP-O towards protection of beta cells that is consistent with the language in our manuscript that was down-toned according to the reviewer's previous suggestions. With respect to our findings with transplanted human islets, we reiterate the challenges inherent to such model which is dependent on optimal islet transplant numbers with high variability between islets preparations from different donors. Consequently, initial re-establishment of normoglycemia and subsequent development into diabetes in this model can be variable. Despite of this inherent variability the data presented in Figure 3b/c combining four independent experiments provide further support of α -IAPP-O's therapeutic potential to delay the progression of diabetes. We have further amended the corresponding section in order not to overstate our findings:

Discussion (page 13, line 312-315):

«While the mean data in the human islet engrafted models ~~clear showed~~ indicated delayed recurrence of diabetes by α -IAPP-O, there was considerable variation among cohorts likely reflecting differences in human islet donors and preparations and proper adaptation of islets after transplantation.

Line 54 also needs to be rewritten since the wording is incorrect and not what was in the cited review or the references cited there. Amyloid deposits are seen in the majority of patients with T2D, not in “the majority of islets from patients with T2D” as stated.

Response: We have amended the appropriate sentence accordingly which now reads as follows: “

“...and forms islet amyloid found in the majority of people with T2D.”

(formerly: “forms amyloid deposits in a majority of pancreatic islets from T2D patients”)

Reviewer #3 (Remarks to the Author):

Fantastic work, really enjoy reading it. I have no specific comment, but a question: Did you try doing some structural studies? Also, can authors give more details on how they selected the antibodies?

Response: We thank reviewer 3 for the kind words and the time and efforts invested to revise and improve our manuscript.

We agree on the importance of structural studies in order to further the understanding on the exact binding mode of α -IAPP-O to IAPP oligomers. As of date we have not performed such studies but are planning this as future experiments.

The antibody selection criteria included high affinity binding to oligomeric forms IAPP with absent binding to unrelated targets as well as physiological monomeric forms of IAPP (Method Section in main text, page 18, lines 407-408).